# DADP: Domain Adaptive Diffusion Policy

**Pengcheng Wang** [1] [*]   **Qinghang Liu** [1] [2] [*]   **Haotian Lin** [3]
**Yiheng Li** [1]   **Guojian Zhan** [1] [4]   **Masayoshi Tomizuka** [1]   **Yixiao Wang** [1]

## Abstract

Learning domain adaptive policies that can generalize to unseen transition dynamics, remains a fundamental challenge in learning-based control. Substantial progress has been made through domain representation learning to capture domain-specific information, thus enabling domain-aware decision making. We analyze the process of learning domain representations through dynamical prediction and find that selecting contexts adjacent to the current step causes the learned representations to entangle static domain information with varying dynamical properties. Such mixture can confuse the conditioned policy, thereby constraining zero-shot adaptation. To tackle the challenge, we propose **DADP** (**D**omain-**A**daptive **D**iffusion **P**olicy), which achieves robust adaptation through unsupervised disentanglement and domain-aware diffusion injection. First, we introduce Lagged Context Dynamical Prediction, a strategy that conditions future state estimation on a historical offset context; by increasing this temporal gap, we unsupervisedly disentangle static domain representations by filtering out transient properties. Second, we integrate the learned domain representations directly into the generative process by biasing the prior distribution and reformulating the diffusion target. Extensive experiments on challenging benchmarks across locomotion and manipulation demonstrate the superior performance, and the generalizability of DADP over prior methods. More visualization results are available on the website .

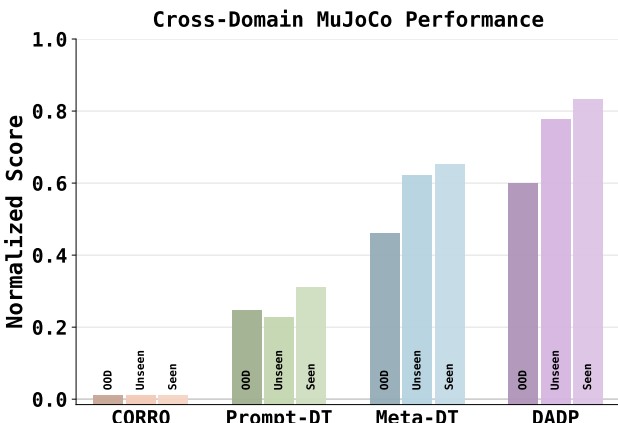

*Figure 1.* Averaged Normalized Performance of baselines across Seen and Unseen settings across all tasks. The results are normalized with random and expert policy performance.

## 1. Introduction

Learning-based polices have achieved remarkable success recently, enabling agents to solve increasingly complex decision-making problems (Liu et al., 2025b; Su et al., 2025). Despite these advances, most existing approaches remain coupled to a specific environment or operating condition (Wang et al., 2024a; Barreto et al., 2017), and their performance often degrades when deployed in the unseen domains (He et al., 2025), which limits the practical applicability of learning-based policies . This mismatch highlights a fundamental challenge: designing a single policy that can generalize efficiently and robustly across domains remains critically important but inherently difficult.

Most prior approaches begin by extracting domain information and leveraging it for decision-making, namely domain representation learning and representation utilization. Regarding the former, many methods extract representations through contrastive learning (Yuan & Lu, 2022; Wen et al., 2024), whose performance often depends on carefully designed objectives and extra data generation. Some other approaches instead employ dynamical prediction as an auxiliary task to implicitly learn the domain representations from transitional dynamics (Lee et al., 2020; Evans et al., 2022); however, the resulting representations are frequently of limited quality due to entangling the static domain information with varying dynamical properties.

---

[*] equal contribution, order is decided randomly [1]University of California, Berkeley, California, USA [2]Peking University, Beijing, China [3]CMU, Pennsylvania, USA [4]Tsinghua University, Beijing, China. Correspondence to: Pengcheng Wang <wangpc@berkeley.edu>, Qinghang Liu <qinghangliu@stu.pku.edu.cn>, Yixiao Wang <yixiao_wang@berkeley.edu>.

Regarding the latter, most methods utilize representations through input concatenation (Kumar et al., 2021; 2022),namely using representations as extra network input, or rely on sequence-modeling architectures (Wang et al., 2024b; Ota, 2024; Huang et al., 2024) to capture domain information in an implicit, end-to-end manner. Such approaches often fail to fully leverage the learned representations, resulting degraded performance.

To tackle these challenges, we propose **DADP** (**D**omain-**A**daptive **D**iffusion **P**olicy), which achieves robust adaptation through unsupervised disentanglement and domain-aware diffusion injection. First, to remove disentangle time-varying properties from the unsupervisedly learned representation, we introduce Lagged Context Dynamical Prediction. Specifically, we break the temporal correlation between the context and the current step by introducing a large historical offset $\Delta t \to \infty$, preventing time-varying information in the context from assisting dynamical prediction and thereby excluding it from the extracted representation during learning. Second, we inject the learned representations directly into the generative process. Specifically, we start denoising with a representation-biased mixed guassian distribution, and reformulate the diffusion target to include the learned representation. We evaluate DADP across locomotion and manipulation tasks across MuJoCo and Adroit, showing consistently superior performance in both modalities capturing ability and generalizability.

In summary, our main contributions are as follows:

- **Unsupervised Representation Learning.** We propose Lagged Context Dynamical Prediction, a simple yet effective approach for unsupervisedly learning domain representations from dynamical prediction.
- **Denoising with Representation-Prediction.** Instead of conditioning, we utilize the domain representations by biasing the prior distribution and reformulating the diffusion target, enabling better policy performance.
- **Superior Performance.** We evaluate DADP on a broader and more challenging set of domain adaption tasks, demonstrating consistently superior performance in domain-adaptivity under the zero-shot setting.
- **Open-sourced Dataset and Pipeline.** We release a complete open-sourced codebase, including the algorithm, datasets, and data generation pipeline, allowing the community to easily customize our framework.

## 2. Related Work

### 2.1. Domain Adaptive Policy

**Sequential Modeling Policy.** Regarding policy architecture, early efforts extend the observation to context composed of multiple consecutive state (Kumar et al., 2021) to provide sufficient information for domain adapation. Subsequent works leverage mature sequence-to-sequence models to better utilize the contextual information, such as Transformers (Chen et al., 2021; Wang et al., 2024b) or Mamba (Ota, 2024; Huang et al., 2024). Among them, Locoformer (Liu et al., 2025b) employs Transformer-XL (Dai et al., 2019) to enable information sharing across episodes and online improvement. However, these methods often rely on purely end-to-end learning, where the absence of intermediate supervision prevents the models from effectively exploiting the implicit dynamical information (Dai et al., 2019).

**Meta RL and In-Context RL.** Regarding algorithms, In Context Reinforcement Learning (ICRL) (Laskin et al., 2022) and Meta Reinforcement Learning (Duan et al., 2016) methods constitute a widely adopted approach, designed to operate over a MDP set by learning from task-level variations during training. In-context Q Learning (Liu et al., 2025a) feed the task representation into a causal transformer with value and policy head for efficient learning across domains. This reflects the dominant approach adopted by most prior works (Kumar et al., 2021; Yuan & Lu, 2022) on ICRL, where the learned representations are provided as additional observation to enable domain-aware decision making. Recent extensions explore sim-to-real co-training (Cheng et al., 2026) and parameter-space skill composition (Liu et al., 2025c), yet remain within the input-conditioning paradigm. Closest to our work, MetaDiffuser (Ni et al., 2023) also incoperates learned domain representations into the diffusion process. However, its representations suffer from the entangled time-varying information due to its representation learning pipeline. As MetaDiffuser is not open-sourced, a direct empirical comparison is not feasible; instead, we compare with it implicitly via ablations on the core differences: (i) representation learning, where MetaDiffuser adopts $\Delta t = 1$ while DADP uses $\Delta t \to \infty$, and (ii) representation utilization, where MetaDiffuser conditions on the representation in the policy input while DADP injects it into the prior distribution. The benefits of both choices are supported by our ablations (Tables 2, 6, 3). Other diffusion-based methods address cross-embodiment transfer via human demonstrations (Pace et al., 2025) or cross-domain editing (Niu et al., 2024), complementary the scope to DADP.

### 2.2. Domain Representation Learning.

Domain refers to the environment's transition dynamics. In practice, it is often characterized by a low-dimensional parameter vector, including environmental parameters (e.g., gravity, friction) or agent-specific parameters (e.g., joint torques, limb lengths). To address the problem of domain adaptive policy learning, many prior works focused on domain representation learning, where a compact representation of domain is inferred from a trajectory of interactions.

**Supervised.** Some works adopt a supervised learning setting, assuming that each domain can be characterized by a low-dimentional accessible environmental factor (Zhang et al., 2025; Lyu et al., 2025), which naturally serves as the target for representation learning. One of the most well-known works is RMA (Kumar et al., 2021; 2022), which achieves online adaptation to different environments by co-training a factor-supervised context encoder and an representation-conditioned policy.

**Unsupervised.** Many other prior works focus on the unsupervised setting, where such environmental factors are assumed to be unavailable. One line of work leverages classical unsupervised learning techniques to cluster data of different domains, such as contrastive learning–based approaches (Li et al., 2020; Wang et al., 2023). Among them, CORRO (Yuan & Lu, 2022) proposes a contrastive learning framework for robust task representations under distribution shifts between training and test be-havior policies. However, such methods often rely on, yet fail to fully exploit, the temporal and sequential structure inherent in control tasks, and are highly dependent on the quality of the extra data generation model. In contrast, our approach is derived from dynamics prediction formulated through sequence modeling, enabling a simpler and more effective capture of domain information.

Another line of work learn domain representations implicitly by introducing dynamics prediction as an auxiliary task, like CaDM (Lee et al., 2020), IIDA (Evans et al., 2022) and CARoL(Hu et al., 2025). However, such methods often suffer from poor representation quality, as they also fail to properly remove the varying information present in the context. In contrast, our method breaks such time-local cues by reconstructing prediction pairs, thereby yielding representations that serve as domain-specific static representations.

## 3. Preliminaries

### 3.1. Problem Formulation

In this work, we formulate the domain adaptive policy learning as an offline meta-RL problem. Specifically, we consider a task set $\mathcal{T} = \{\mathcal{T}_i\}_{i=1}^n$, where each task $\mathcal{T}_i$ consists of an Markov Decision Process (MDP) and a policy that has been pre-trained on this MDP.

$$\mathcal{T} = \{\mathcal{T}_i\}_{i=1}^n = \{(\mathcal{M}_i, \pi_i)\}_{i=1}^n \qquad (1)$$

The MDP can be defined by a tuple $\mathcal{M}_i = (\mathcal{S}, \mathcal{A}, R, p_i)$, where $\mathcal{S} \subseteq \mathbb{R}^n$ is a continuous state space, $\mathcal{A} \subseteq \mathbb{R}^m$ is a continuous action space, $R : \mathcal{S} \times \mathcal{A} \to \mathbb{R}$ is the reward function, and $p_i : \mathcal{S} \times \mathcal{A} \times \mathcal{S} \to [0, \infty)$ is the transition probability function. Across all MDPs, the state space $\mathcal{S}$, action space $\mathcal{A}$, and reward function $R$ are shared, while the transition dynamics $p_i$ differ across tasks, i.e., $p_i \neq p_j$. We emphasize

that in our setting "cross-domain" does not mean "cross-task": all domains share the same task type $(\mathcal{S}, \mathcal{A}, R)$ and differ only in their transition dynamics. Although DADP's mechanisms do not inherently require shared rewards (see Section 5.3.1 and Table 6), we restrict the current scope to dynamics variation as a deliberate choice, since it directly relates to practical robotics challenges such as sim-to-real transfer and cross-embodiment deployment.

For each task $\mathcal{T}_i$, the agent is given an offline dataset $\mathcal{D}_i$, collected by executing a *domain-specific* expert policy $\pi_i$ in the corresponding environment $\mathcal{M}_i$. The expert policy $\pi_i$ is constructed by training a reinforcement learning (RL) agent to (near-)optimality on $\mathcal{T}_i$, and is therefore specialized to the dynamics and reward of $\mathcal{M}_i$. Our objective is to learn a policy $\pi$ using the datasets $\{\mathcal{D}_i\}$ from the training task set $\mathcal{T}^{\text{train}}$, and to maximize discounted return for all tasks in $\mathcal{T}^{\text{train}} \cup \mathcal{T}^{\text{test}}$, i.e., $J(\pi) = \mathbb{E}_\pi[\sum_{t=0}^\infty \gamma^t R(s_t, a_t, s_{t+1})]$, where $\gamma \in (0, 1)$ is the discount factor.

### 3.2. Diffusion Policy

Diffusion policies (Chi et al., 2025; Ho et al., 2020) model the action generation as a stochastic denoising process conditioned on the observatoin. Specifically, a diffusion policy learns a conditional action distribution $q(a \mid s)$ through a predefined forward diffusion process and a learned reverse denoising process. Throughout the paper we use $k \in \{1, \dots, K\}$ to index the diffusion step (reserving $t$ for the environment timestep). The forward process gradually perturbs a clean action $a^0$ into noisy latent variables $a^k$ by

$$q(a^k \mid a^{k-1}) = \mathcal{N}(a^k \mid \sqrt{\bar{\alpha}_k}\, a^{k-1}, \Sigma_k), \qquad (2)$$

where $\mathcal{N}(\mu, \Sigma)$ denotes a Gaussian distribution with mean $\mu$ and covariance $\Sigma$, $\bar{\alpha}_k \in \mathbb{R}$ is the variance schedule, $\Sigma_k$ is the per-step covariance, and $a^0 \sim q(a \mid s)$ is an action sampled from the data distribution.

Starting from Gaussian noise, actions are generated by iteratively applying the learned reverse process. The denoising policy $\epsilon_\theta(a^k, s, k)$ predicts the noise at each diffusion step conditioned on input. The policy is trained by minimizing a simplified surrogate objective (Ho et al., 2020):

$$\mathcal{L}_{\text{diff}}(\theta) = \mathbb{E}_{a^0, \epsilon, k}[\|\epsilon - \epsilon_\theta(a^k, s, k)\|^2], \qquad (3)$$

where $a^k = \sqrt{\bar{\alpha}_k}\, a^0 + \sqrt{1 - \bar{\alpha}_k}\, \epsilon, \epsilon \sim \mathcal{N}(0, I)$. The objective encourages the model to recover the injected noise at each diffusion step. At inference time, the policy samples an action by initializing from Gaussian noise and iteratively applying the learned denoising model conditioned on the current state to get the clean action distribution.

## 4. Domain Adaptive Diffusion Policy

### 4.1. Learn Representation by Extracting Static Info

To enable the policy to possess domain adaptive capability, we firstly train a context encoder $E_\phi(\cdot)$ to learn an effective domain representation $z_t$ from the context $\tau_t$. We choose to learn the representation from context since (i) the domain factors that govern dynamics are typically latent and must be inferred from observed transitions, and (ii) at test time, the agent generally has access only to interaction history rather than privileged environment parameters.

In this work, we learn the representation implicilty from dynamical prediction. Typically, the context as encoder input is from the most recent history (Ni et al., 2023):

$$z_t = E_\phi(\tau_t); \hat{s}_{t+1} = f_\theta(s_t, a_t, z_t);$$
$$\tau_t = (s_{t-H}, a_{t-H}, \ldots, s_{t-1}, a_{t-1}). \tag{4}$$

This context selection is intuitive, as it aligns with the usage in online policy inference, where the most recent history is used as context. Note that there are two types of necessary information that can be infered from the context, which are both necessary for accurate next state prediction: static information $\xi$ that represents the domain-specific dynamics (e.g. gravity), varying information $\omega_t$ that includes instantaneous dynamical properties not captured in the current state (e.g. higher-order temporal derivatives of states):

$$s_{t+1} = f(s_t, a_t, \xi, \omega_t); \ z_t = E_\phi(\tau_t) = (\xi^z, \omega_t^z), \tag{5}$$

where $f$ represents the ground-truth forward dynamics, $\xi^z$ and $\omega_t^z$ represent the inferred information from the context.

Note that since the context is drawn from the most recent history, the inferred varying information $\omega_t^z$ is temporally aligned with the ground-truth varying information $\omega_t$ required for prediction task:

$$z_t = \arg\min_{z=(\omega_t^z, \xi^z)} \mathbb{E}_\mathcal{D} \|s_{t+1} - \hat{s_{t+1}}\|^2. \tag{6}$$

As a result, $z_t = (\xi, \omega_t)$ becomes a natural global minimum for the dynamical prediction task.

However, recall that a domain corresponds to the environment's transition dynamics, usually parameterized by a low-dimensional vector and therefore inherently static. For domain adaptation, the primary purpose of the representation is to provide a stable descriptor of the domain: it should reflect the persistent domain factors $\xi$ across time, while remaining insensitive to ephemeral variations $\omega_t$ that are not stable within a domain. Encoding $\omega_t$ into $z_t$ can cause *representation drift* within the same domain, reducing separability across domains and harming generalization when

$z_t$ is used as a domain descriptor for downstream policy learning. This motivates us to seek a mechanism for disentangling time-varying information $\omega_t$ from $z_t$.

To remove $\omega_t$, we propose Lagged Context Dynamical Prediction. Specifically, we introduce a temporal offset $\Delta t$ to weaken the contribution of time-local cues in the context for next state prediction. With $\Delta t$, we can adjust the "distance" between the context and the current timestep:

$$z_{t-\Delta t} = E_\phi(\tau_{t-\Delta t}); \hat{s}_{t+1} = f_\theta(s_t, a_t, z_{t-\Delta t});$$
$$\tau_{t-\Delta t} = (s_{t-H+1-\Delta t}, a_{t-H+1-\Delta t}, \ldots, s_{t-\Delta t}, a_{t-\Delta t}), \tag{7}$$

, where the original context corresponds to $\Delta t = 1$.

From an information-theoretic viewpoint, as $\Delta t$ increases, the offset context becomes less informative about instantaneous variations. Since $z_{t-\Delta t}$ is a function of $\tau_{t-\Delta t}$, we have $I(\omega_t; z_{t-\Delta t} \mid s_t, a_t, \xi) \leq I(\omega_t; \tau_{t-\Delta t} \mid s_t, a_t, \xi)$. In this way, $z_{t-\Delta t}$ discards the instantaneous variations , while $\xi$ remains informative since it is time-invariant within a domain. Consequently, optimizing prediction with offset contexts biases the representation toward the static domain factors $\xi$ rather than transient $\omega_t$.

When $\Delta t \to \infty$, the representation $z_{-\infty} = (\xi, \overline{\omega})$ becomes static, where $\xi$ is the static domain information, $\overline{\omega}$ is a averaged varying properties that minimizes the prediction loss on the dataset distribution. This can be easily achieved by selecting the context from another episode in the same domain. Please refer to Appendix A for a toycase explanation.

Throughout this work, we adopt the universal default $\Delta t \to \infty$ (implemented by sampling the context from another episode in the same domain) consistently across all tasks and environments. This choice is parameter-free: it is theoretically grounded as the limit that retains only static, time-invariant information, and our empirical results (Table 2 across all four MuJoCo environments and Table 3, together with the extended utilization ablation in Appendix C.7) show monotonic improvement in both representation quality and downstream performance as $\Delta t$ increases. As a result, no per-task tuning of $\Delta t$ is required, including for tasks with very different dynamical scales (e.g. high-speed locomotion vs. contact-rich manipulation).

### 4.2. Utilize Representation by Diffusion Modulation

With good domain representations, it remains to determine how to better utilize them to enable domain-aware decision-making. In this work, we build our method upon diffusion policy, as it has been widely adopted and has demonstrated strong performance across various control tasks.

In the standard Diffusion Policy (Chi et al., 2025), the denoising process starts from pure Gaussian noise, where different denoising trajectories are governed by single

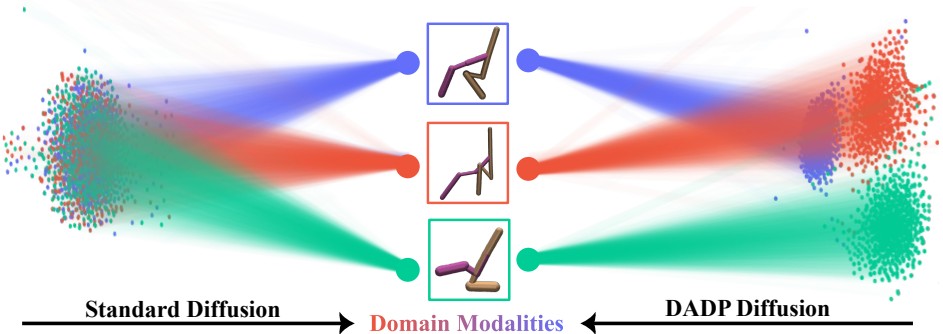

*Figure 2.* t-SNE Visualization of Denoising Process of Standard Diffusion and DADP. The sampled points from prior distribution and utilized representation are contructed from the training datasets and learned context encoder.

representation-conditioned policy:

$$a^k = \sqrt{\bar{\alpha}_k}\, a^0 + \sqrt{1 - \bar{\alpha}_k}\, \epsilon. \tag{8}$$

If we simply take the learned representation as extra policy inputs, the diffusion policy has to reconstruct different domain-specific action modalities from every sampled point in the prior gaussian distribution equally. This entanglement leads to mixed denoising trajectories in the latent noise space, preventing the policy from exploiting the structure of the noise to better leverage domain information, and consequently resulting in degraded performance.

To solve this challenge, intead of utilizing the representation as condition, we inject the representation into the generation. Specifically, DADP initializes the denoising process from a Gaussian Mixture by incorporating the learned representation $z$ into the forward process. Following the formulation for structured diffusion models (e.g. Mixed DDIM (Jia et al., 2024)), we define the perturbed action $a^k$ at step $k$:

$$a^k = \sqrt{\bar{\alpha}_k}\,(a^0 - z) + z + \sqrt{1 - \bar{\alpha}_k}\, \epsilon, \tag{9}$$

where $z$ is the learned domain representation obtained in Section 5.3.1 (i.e. $z = z_{t-\Delta t}$, and $z_{-\infty}$ for our universal default $\Delta t \to \infty$). At the final diffusion step $K$ we have $a^K = z + \epsilon$, a Gaussian centered at the domain-specific representation $z$. Since different domains have distinct $z$ values that form well-separated clusters (Figures 3 and 6), the marginal prior aggregated over all domains forms a *mixed Gaussian* with one peak per domain. Here, "domain-specific action modality" refers to the optimal action distribution under one set of dynamics — not a new task: all domains share the same task type $(\mathcal{S}, \mathcal{A}, R)$, but optimal actions differ because dynamics differ. In this way, we inject the domain information into the prior distribution as shown in Figure 2.

By rearranging Eq. (9), the clean action $a^0$ can be estimated from $a^k$ and the predicted noise $\epsilon^\theta$:

$$a^0 = \frac{a^k - z - \sqrt{1 - \bar{\alpha}_k}\, \epsilon^\theta}{\sqrt{\bar{\alpha}_k}} + z. \tag{10}$$

Utilizing the Denoising Diffusion Implicit Model (DDIM) formulation (Song et al., 2022) and omitting the stochastic noise injection for clarity, the reverse step is defined as:

$$a^{k-1} = \sqrt{\bar{\alpha}_{k-1}}\, a^0 + \sqrt{1 - \bar{\alpha}_{k-1}}\, \frac{a^k - \sqrt{\bar{\alpha}_k}\, a^0}{\sqrt{1 - \bar{\alpha}_k}}. \tag{11}$$

Substituting Eq. (10) into Eq. (11) yields:

$$\begin{aligned}
a^{k-1} = & \sqrt{\frac{\bar{\alpha}_{k-1}}{\bar{\alpha}_k}}\left(a^k - z - \sqrt{1 - \bar{\alpha}_k}\, \epsilon^\theta\right) + \sqrt{\bar{\alpha}_{k-1}}\, z \\
& + \frac{\sqrt{1 - \bar{\alpha}_{k-1}}}{\sqrt{1 - \bar{\alpha}_k}}\left(z + \sqrt{1 - \bar{\alpha}_k}\, \epsilon^\theta - \sqrt{\bar{\alpha}_k}\, z\right).
\end{aligned} \tag{12}$$

In this work, instead of setting $\epsilon^\theta$ as the prediction target as usual, we propose a joint prediction objective, where the model learns to predict a composite term $\hat{\epsilon}^\theta$ representing the noise and the representation shift together:

$$a^k = \sqrt{\bar{\alpha}_k}\, a^0 + \underbrace{\left(1 - \sqrt{\bar{\alpha}_k}\right) z + \sqrt{1 - \bar{\alpha}_k}\, \epsilon}_{\hat{\epsilon}^\theta}. \tag{13}$$

Under this scheme, the sampling iteration simplifies as:

$$a^{k-1} = \sqrt{\frac{\bar{\alpha}_{k-1}}{\bar{\alpha}_k}}\left(a^k - \hat{\epsilon}^\theta\right) + \frac{\sqrt{1 - \bar{\alpha}_{k-1}}}{\sqrt{1 - \bar{\alpha}_k}}\hat{\epsilon}^\theta \tag{14}$$

In this way, we not only bias the prior distribution, but also introduce extra supervision on each denoising steps to further guide and simplify the denoising process. A complete empirical analysis of these variants can be found in Section 5.3.2, which shows the great policy performance gain of the proposed approach.

# 5. Experiments

With experiments, we aim to answer these questions:

1. How does the performance of proposed DADP policy compared to existing SOTA methods?
2. Does the proposed Lagged Context Dynamical Prediction contributes to the representation quality?
3. Does the proposed representation utilization further improve the performance of the diffusion policy?

## 5.1. Experimental Setup

**Environments.** Previous evaluations of domain adaptation policies have largely focused on existing locomotion settings (Todorov et al., 2012; Ni et al., 2023), where domain randomization typically is restricted to mild variations (e.g., friction or gravity shifts), which tend to have limited impact on the optimal gait. In this work, we expand the locomotion tasks to four environments and further introduce morphological variations. As a result, the gaits across different domains exhibit greater diversity compared to prior works. Please refer to Appendix D.2 for the dataset visualization. Furthermore, to demonstrate the generality and applicability of DADP in environments with complex dynamics, we additionally incorporate a manipulation benchmark, Adroit (Rajeswaran et al., 2017), into our experiments.

**Data Generation.** For locomotion environments, we follow the data collection pipeline of CORRO (Yuan & Lu, 2022), constructing the task set by sampling different environmental factors in the parametric space. For each task, we use SAC (Haarnoja et al., 2018) to train a task-specifc expert for offline data collection, which contains 25 domains. For manipulation tasks in Adroit, we adopt the pre-collected dataset from ODRL (Lyu et al., 2024), which contains 3 domains. Please refer to Appendix B.1 for more details.

**Baselines.** We consider the following methods as our baselines. Please refer to Appendix B.2 for more details.

- **CORRO** (Yuan & Lu, 2022) proposes a contrastive learning framework for robust task representations under distribution shifts, outperforming prior context-conditioned policy-based methods.
- **Prompt-DT** (Xu et al., 2022) leverages Transformer-based sequence modeling with a prompt formulation to enable few-shot adaptation in offline RL, serving as a strong end-to-end meta-RL baseline.
- **Meta-DT** (Wang et al., 2024b) incorporates an additional learned domain representation as an augmented observation, further improving performance and representing a SOTA baseline in domain adaptation task.

**Training and Evaluation.** Training is conducted in two stages. Firstly, a context encoder is pre-trained on training dataset to extract domain representations from trajectories; secondly, a diffusion policy is trained with the fixed learned context encoder across 5 random seeds.

During evaluation, we test the policies with zero-shot setting, where the contexts are online collected during policy rollout. Compared to the few-shot setting, which assumes access to expert datasets from unseen domains as context, the zero-shot setting more closely reflects practical deployment scenarios (Liu et al., 2025b). We evaluate all the baselines under three settings: **Seen**, **Unseen**, and **OOD**. The **Seen** setting measures performance on the domains present in the training dataset, assessing the policy's ability to master multiple training domains. The **Unseen** setting samples 5 new parameter combinations from *within* the training factor space (i.e., novel domains whose factors interpolate between training values), evaluating in-support generalization. The **OOD** setting samples 5 parameter combinations from ranges that lie *outside* the training factor space, probing genuine out-of-support extrapolation; the exact out-of-support ranges per environment are listed in Appendix C.4. For the Adroit benchmark, we instead use the Easy and Hard domains as the Seen setting and the Medium domain as the Unseen setting; OOD is not available for Adroit due to dataset constraints. Please refer to Appendix E for more details.

## 5.2. Experimental Results

As shown in Table 1, across all evaluated environments, DADP consistently achieves strong performance under all three settings (Seen, Unseen, and OOD), outperforming or matching the best-performing baselines in nearly all cases, with the advantage being most pronounced under OOD. These results indicate that DADP effectively captures and leverages the domain information to generalize across both in-support novel domains and genuine out-of-support extrapolation. Moreover, across all environments, DADP achieves stable performance with smaller standard deviation across seeds, demonstrating its strong stability and practicability.

We additionally evaluate DADP under two practically important regimes. (i) *Non-stationary dynamics*: although DADP is trained under stationary dynamics, the encoder re-estimates the domain representation from the most recent online context, and the same checkpoint remains performant across all four Walker2d friction-variation schedules (Appendix C.5). (ii) *Inference efficiency*: the representation-biased prior shifts the diffusion start point closer to the target action manifold, making the generation tolerant to aggressive step reduction — under one-step DDIM, a standard diffusion policy collapses while DADP retains a substantial fraction of its performance (Appendix C.6). These results suggest DADP is a practical fit for real-time control and deployments where dynamics may evolve at test time.

*Table 1.* Benchmark performance across different environments under **Seen** (training domains), **Unseen** (new parameter combinations sampled within the training factor space), and **OOD** (parameters sampled *outside* the training factor space; only available for MuJoCo) settings. Parameter ranges for the OOD setting are listed in Appendix C.4. The results are from the last checkpoint, presented in mean $\pm$ std across 5 random seeds, where the highest mean performance of each variant is bolded, and the second highest is underlined.

| Environment | Setting | Expert | CORRO | Prompt-DT | Meta-DT | DADP (Ours) |
|---|---|---|---|---|---|---|
| **HalfCheetah** | Seen | 4575 | -301±42 | 1640±194 | 3857±234 | **3978**±66 |
| | Unseen | – | -246±36 | 250±375 | **3174**±501 | 3001±225 |
| | OOD | – | -293±67 | 733±125 | 2776±380 | **3371**±257 |
| **Walker2d** | Seen | 7101 | 61±49 | 590±57 | 1304±586 | **3999**±174 |
| | Unseen | – | 66±69 | 435±157 | 889±579 | **2834**±285 |
| | OOD | – | 9±28 | 427±93 | 954±252 | **2197**±173 |
| **Ant** | Seen | 3598 | -867±430 | 700±189 | 3045±128 | **3052**±30 |
| | Unseen | – | -962±553 | 208±126 | 3187±899 | **3485**±83 |
| | OOD | – | -1177±567 | 353±138 | 1498±1184 | **1903**±64 |
| **Hopper** | Seen | 1555 | 80±11 | 935±65 | 1140±156 | **1631**±47 |
| | Unseen | – | 61±21 | 1148±150 | 1208±99 | **1686**±47 |
| | OOD | – | 67±29 | 1048±51 | 1070±180 | **1271**±48 |
| **Door** | Seen | 3233 | -50±13 | **2116**±177 | 1283±323 | 1428±44 |
| | Unseen | 3261 | -58±2 | 1080±209 | 1294±228 | **1494**±81 |
| **Relocate** | Seen | -1.92 | -12.3±1.92 | -7.44±0.10 | -6.06±0.40 | **-5.81**±0.15 |
| | Unseen | -1.70 | -12.0±0.72 | -6.47±0.36 | -5.77±0.42 | **-5.74**±0.15 |

## 5.3. Ablation Study

In the ablation study, we aim to examine how each proposed component of our diffusion policy contributes to the overall performance. We focus on the two tasks where DADP achieves the different level of gains over the other baselines in the main results, Walker2d and HalfCheetah. Please refer to Appedix C for additional experiments and analysis.

### 5.3.1. EFFECT OF $\Delta t$ ON REPRESENTATION QUALITIES

To evaluate the representation qualities, we apply the learned context encoder to encode the training dataset, obtaining the domain representation set. Specifically, we use the linear probe accuracy (Oord et al., 2018) and reconstruction loss to evaluate the representation qualities. For linear probe accuracy, we train a single-layer softmax linear classifier to predict the one-hot domain index corresponding to each representation; for reconstruction loss, we train a two-layer MLP to predict the exact dynamical parameter vectors. Furthermore, we treat the supervised representation as an upper bound on performance, as it is trained with access to ground-truth labels—specifically in our setting, the environment parameters and the corresponding one-hot domain indices.

As shown in Table 2, as $\Delta t$ increases, both metrics consistently improve across all four MuJoCo environments, eventually comparable to that of supervised representations. This indicates that larger $\Delta t$ effectively breaks time-local cues,

resulting in representations with stronger representative capacity for domain classification and encoding underlying parameters. Moreover, we can explain the larger performance gain on Walker2d compared with HalfCheetah, since the former representation benefits more from larger $\Delta t$.

Among the four MuJoCo environments, Hopper is the only one whose dataset does not include morphological variations (see Appendix B.1); its domains differ only via minor friction and damping, producing action patterns across domains that are inherently hard for *any* unsupervised encoder to separate. This is reflected in Hopper's lower probe accuracy at $\Delta t \to \infty$ compared with the other three environments. Despite this, the monotonic improvement still holds across both metrics on Hopper, showing that the lagged-context technique still extracts an improving signal even when the dataset offers minimal across-domain separability. An interesting takeaway for cross-domain dataset design follows naturally: the $\Delta t \to \infty$ probe accuracy can serve as a fully unsupervised diagnostic of action-pattern diversity in a cross-domain dataset — high values flag richly distinguishable domain-specific behaviors (as in Walker2d, HalfCheetah, and Ant under morphological variation), whereas low values indicate datasets where domains are inherently hard to separate from behavior alone as in Hopper.

We also provide the t-SNE visualizations of the representations of Walker2d learned with different $\Delta t$ in Figure 3. As $\Delta t$ increases, the representations from different domains

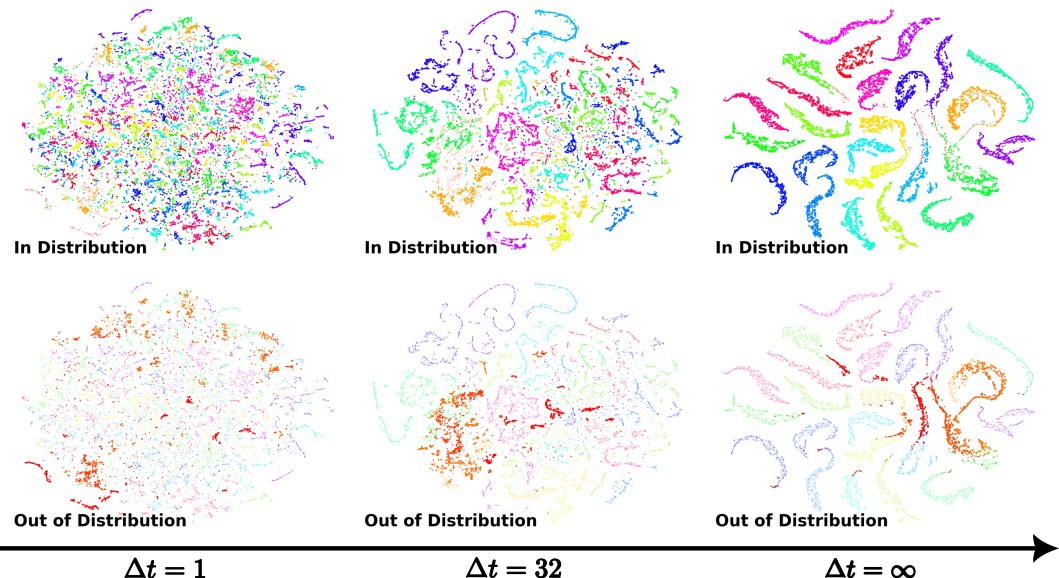

*Figure 3.* t-SNE visualization of walker representations learned with different $\Delta t$.

*Table 2.* Normalized Representative Metrics of Learned Embeddings with different $\Delta t$, evaluated on all four MuJoCo environments.

| Environment | Metrics | $\Delta t = 1$ | $\Delta t = 4$ | $\Delta t = 16$ | $\Delta t = 32$ | $\Delta t = \infty$ | Supervised |
|---|---|---|---|---|---|---|---|
| **Walker2d** | Linear Probe Accuracy | 27.9% | 35.7% | 48.5% | 64.9% | **99.3%** | 99.8% |
| | Reconstruction Loss | 476.1 | 427.5 | 312.6 | 229.0 | **3.2** | 1.0 |
| **HalfCheetah** | Linear Probe Accuracy | 68.6% | 84.7% | 96.7% | 98.3% | **99.9%** | 99.9% |
| | Reconstruction Loss | 45.9 | 19.8 | 3.4 | 1.1 | **0.4** | 1.0 |
| **Ant** | Linear Probe Accuracy | 62.4% | 98.5% | 99.6% | 99.8% | **99.8%** | 99.8% |
| | Reconstruction Loss | 485.1 | 34.6 | 2.2 | 2.2 | **1.8** | 1.0 |
| **Hopper** | Linear Probe Accuracy | 9.0% | 11.2% | 11.8% | 15.5% | **26.4%** | 99.0% |
| | Reconstruction Loss | 28.9 | 28.7 | 27.5 | 26.8 | **21.1** | 1.0 |

gradually distinctly cluster. Compared some previous methods (Yuan & Lu, 2022; Li et al., 2020) based on contrastive learning, our approach can achieve great embedding qualities from a simple objective without extra data generation.

Furthermore, we validate that the static representation can lead to better policy performance. As shown in Table. 3, condional policy can benefit from the higher-quality representations. With the proposed utilization, the performance gain can achieve comparable performance with Supervised baseline whose representation is trained with supervised learning as in Sec. 5.3.1.

We also validate that the performance gain can be consistently achieved across reward-changing environments and different datasets with a similar ablation applied to Meta-DT, where introducing larger $\Delta t$ yields consistent gains across both domain-adaptive and the majority of reward-changing environments; full results are reported in Appendix C.1.

### 5.3.2. EFFECT OF REPRESENTATION UTILIZATION

In this section, we investigate how different representation utilization affect the resulting policy performance. Specifically, we mainly focues on the following utilizations:

1. **Null.:** We remove the representation in policy input and generation, serving as an end-to-end baseline.
2. **Cond.:** We utilize the representation as the extra input to the policy for conditional generation.
3. **w/o Predict:** We utilize the representation to bias the prior distribution to a mixed guassian.
4. **w/ Predict:** Upon w/o Predict, we further utilize the representation as part of policy prediction target.

As shown in Table 3, DADP achieve superior performance across all the variants. This indicates that our proposed utilization strategy maximizes the effectiveness of the learned

*Table 3.* Ablation Results on Representation Utilization.

| Variants | Options | | Walker2d | | | HalfCheetah | | |
|---|---|---|---|---|---|---|---|---|
| | Representation | Utilization | Seen | Unseen | Mastery | Seen | Unseen | Mastery |
| End-to-End Diffusion | Null. | Null. | 3722 | 2852 | 40% | 3509 | 2496 | 80% |
| Conditional Policy | $\Delta t = 1$ | Cond. | 2093 | 1617 | 0% | 3740 | 2594 | 80% |
| Better Representation | $\Delta t = \infty$ | Cond. | 3394 | 1813 | 28% | 3603 | 2744 | 76% |
| Mixed DDIM | $\Delta t = \infty$ | w/o Predict | 3356 | 1908 | 36% | 3533 | 3012 | 84% |
| **DADP (Ours)** | $\Delta t = \infty$ | w/ Predict | **3991** | **3015** | **44**% | **4100** | **3055** | **96**% |
| Expert | Null. | Null. | 7101 | - | 100% | 4575 | - | 100% |
| Supervised | Supervised | w/ Predict | 4014 | 2540 | 44% | 3846 | 3152 | 88% |

high-quality representations. The same qualitative trends hold on Ant and Hopper (Appendix C.7, Table 11), confirming that the utilization findings generalize across all four MuJoCo locomotion environments.

To validate the source of performance gain, we introudce a new metric, *mastery*, which is ratio of Seen domains that policy achieve 60% of the expert policy performance. Across all the variants, DADP achieves the highest mastery, showing its strong capability to master diverse domains and locate the target manifold of the corresponding domain, resulting better mastery across the domains. In real-world application, higher mastery implies that a single policy can adapt to a broader range of embodiments and diverse environments. Please refer to Appedix D.2 for visualization results.

To visualize if the representation-prediction utilization can enable effective denoising process, we first rollout the corresponding variants in a specific domain. Next, we split the trajectories into contexts, apply the learned context encoder and visualize the representations of the resulting trajectories. As shown in Figure 4, compared with Mixed DDIM and Conditional Policy, DADP representation accurately locate the running policy to the target domain thus better leverage the in-distribution capability for better control performance.

Another interesting obeservation is, despite its simplicity, End-to-End Diffusion outperforms many variants. This suggests that diffusion-based policies constitute a particularly well-suited policy architecture for domain adaptation problems, and remain underexplored in this context.

## 6. Conclusion

We propose DADP, a diffusion policy achieves robust domain adaptation through unsupervised disentanglement and domain-aware diffusion injection. To obtain high-quality domain representations unsupervisedly, we propose Lagged Context Dynamical Prediction to remove the time-varying

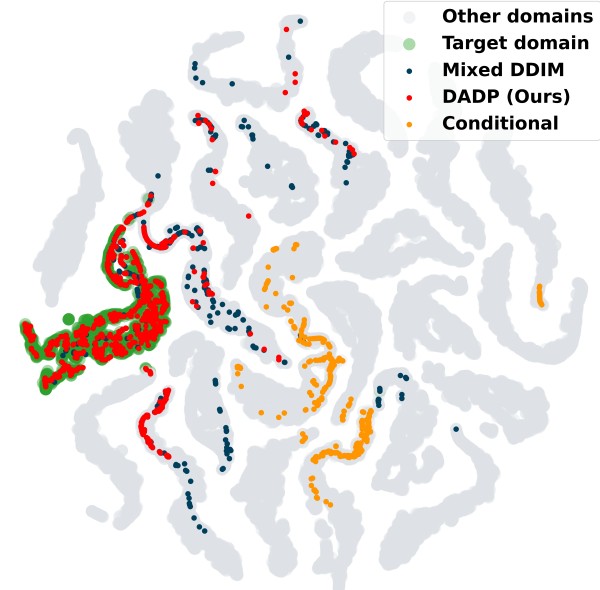

*Figure 4.* Walker Online Adaptation Representation Visualizations

information presents in the context. With the learned representations, we bias the prior distribution and reformulate the diffusion target, achieving SOTA performance and generalizability across diverse challenging benchmarks with verifiable analysis and visualization.

**Limitations.** In this work, we focus on stationary (time-invariant) dynamics and therefore distangle static information from the time-varying information in the context. Nonetheless, time-varying signals can be crucial in non-stationary environments, where they may reflect evolving dynamics that a policy must track for effective control. In future work, we plan to explore how to jointly disentangle and retain the time-varying information, and extend DADP to non-stationary dynamical environments settings.

## Impact Statement

This paper presents work whose goal is to advance the field of Machine Learning. There are many potential societal consequences of our work, none which we feel must be specifically highlighted here.

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

## A. Toycase of $\Delta t$ Design

The toycase can be illustrated with Figure 5. We consider a
toy example of a ball vertical projectile motion under gravity
without air resistancem, whose time-step length $t_0$ is known.
At each time step, the state of the ball is represented solely by
its vertical position $s_t = y_t$, which constitutes an incomplete
state representation without the vertical speed $v_t^y$. Our goal is to
infer the unique scalar environmental factor of this system—the
gravitational acceleration $g$—by predicting the next state:

$$y_{t+1} = y_t + v_t^y t_0 + \frac{1}{2} g t_0^2 \tag{15}$$

Consider the prediction with the most recent context with $\Delta t = 1$, $\tau_{\Delta t=1} = (y_{t-3}, y_{t-2}, y_{t-1})$, which is encoded as $z_{\Delta t=1} = E_\phi(\tau_{\Delta t=1})$. Note that for continuous states of length $L = 3$,
there are always two types of information that can be extracted
from the sequence $(y_{T-2}, y_{T-1}, y_T)$.

- **Static gravity** $g = \frac{1}{t_0^2}(y_T + y_{T-2} - 2y_{T-1})$

- **Varying speed** $v_T^y = \frac{1}{2t_0}(4y_{T-1} - 3y_{T-2} - y_T)$

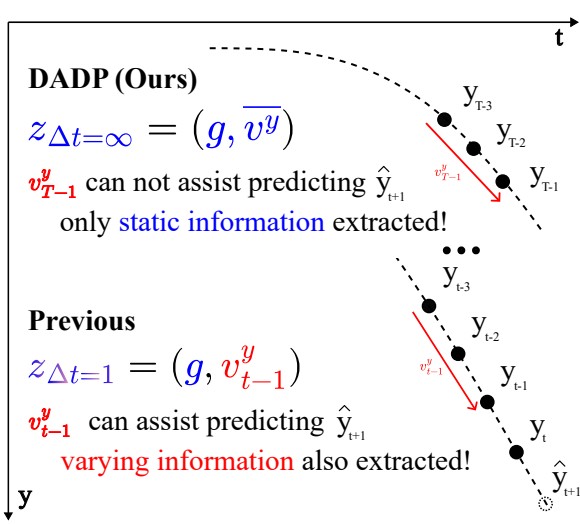

*Figure 5.* Intuition of the $\Delta t$ desing: since the varying velocity
inferred from another episode in the same domain can not
assist the prediction, only static gravity will be extracted in
the representation.

Due to the time-local cues, the extracted velocity from the
context can assist the prediction to achieve lower prediction loss
by extend itself to the current step.

$$v_t^y = v_{t-1}^y + g t_0 = v_T^y + g t_0 \tag{16}$$

We assume that the neural network can eventually achieve zero loss on this prediction; under this assumption, the learned
representation must encode both types of information:

$$z_{\Delta t=1} = \arg \min_{z=(g_z, v_z)} \mathbb{E}\|y_{t+1} - \hat{y_{t+1}}\|^2 = \arg \min_{z=(g_z, v_z)} \mathbb{E}\|g - g_z\|^2 + \|v_t^y - v_z\|^2 = (g, v_{t-1}^y + g t_0) \tag{17}$$

However, this makes the representation a mixture of varying information and static domain representation, while only
the latter is desired for representation learning. To remove the time-varying component, one can reduce the influence of
time-local cues by simply increasing the $\Delta t$, namely the distance between the context and the current step.

Now consider the context from another episode in the same domain $\tau_{\Delta t=\infty} = (y_{t-3}, y_{t-2}, y_{t-1})$. In this case, the varying
speed can not be extended to the current step. To predict next state with lowest loss, the encoder would be enforced to learn
the static information: gravitational acceleration $g$ and the averaged velocity $\overline{v^y} = \mathbb{E}_\mathcal{D}[v_t^y]$:

$$z_{\Delta t=\infty} = \arg \min_{z=(g_z, v_z)} \mathbb{E}\|y_{t+1} - \hat{y_{t+1}}\|^2 = \arg \min_{z=(g_z, v_z)} \mathbb{E}\|g - g_z\|^2 + \|v_t^y - v_z\|^2 = (g, \overline{v^y}) \tag{18}$$

In this case, we recover a static estimate of the gravitational acceleration. Meanwhile, the additionally learned average veloc-
ity term serves as complementary static information that characterizes the overall distribution, enabling the representation to
remain stable within each domain while also capturing salient behavioral patterns.

# B. Implementation Details

## B.1. Expert Dataset

**Dataset Generation.** We generated datasets across four MuJoCo environments: Ant, HalfCheetah, Walker2d, and Hopper. For each environment, we varied specific physical parameters to create diverse dynamics. For Ant, HalfCheetah, and Hopper, the datasets comprise 25 distinct sets of dynamics parameters, with each set containing 100 episodes. Due to the higher complexity of the Walker2d environment, we generated 25 distinct dynamics parameters with 300 episodes per set to ensure sufficient coverage. The maximum episode length is set to 1,000 transitions. Due to its inherent instability of Hopper expert data generation, we limit its transitions to 500 and do not introduce morphological variations. To generate the data, we trained a Soft Actor-Critic (SAC) policy for each parameter setting and collected rollouts from the resulting policies.

For the Adroit manipulation domain, we utilized datasets from (Lyu et al., 2024), specifically selecting the *relocate-shrink-finger* and *door-shrink-finger* tasks. Following the benchmark protocols, we used the "Easy" and "Hard" variants for training. Refer to (Lyu et al., 2024) for the specific configurations of these tasks.

*Table 4.* Dynamics parameter ranges for the MuJoCo environments.

| Environment | Parameter | Range |
|---|---|---|
| **Ant** | Leg Length (4 legs) | $[0.465, 1.612]$ |
| **Hopper** | Joint Damping (3 joints) | $[0.75, 1.19]$ |
| | Friction | $[0.75, 1.19]$ |
| **HalfCheetah** | Back Leg Mass | $[2.99, 6.69]$ |
| | Torso Length | $[0.336, 0.951]$ |
| | Head Length | $[0.084, 0.238]$ |
| | Front Leg Lengths | $[0.075, 0.400]$ |
| **Walker2d** | Friction | $[1.32, 2.28]$ |
| | Torso / Foot Length | $[0.30, 0.67]$ |
| | Thigh / Leg Length | $[0.27, 0.51]$ |
| | Mass (Left Thigh/Leg/Foot) | $[2.99, 6.69]$ |

## B.2. Baselines

We benchmark our method against four baselines representing distinct training paradigms, including an offline RL-based approach, two Decision Transformer (DT)-based approaches, and one diffusion-based approach. For reproducibility, we utilize the official implementations for all baselines, with specific adaptations for our setting as detailed below:

**CORRO** (Yuan & Lu, 2022). We employ the official implementation of this robust offline meta-RL method. It utilizes contrastive learning to acquire robust task representations, generating a latent representation from the task context. The offline RL policy is then conditioned on this representation to handle distribution shifts effectively.

**Prompt-DT** (Xu et al., 2022). Building on the official Decision Transformer implementation, this method utilizes trajectory segments as prompts to encode task information. While the standard inference pipeline samples prompts from an expert dataset, we modify this process to ensure a fair comparison under our zero-shot setting. Specifically, we construct the prompt using the agent's recent interaction history directly gathered from the environment.

**Meta-DT** (Wang et al., 2024b). We utilize the official Meta-DT codebase, which trains an encoder to compress trajectory segments into latent representations. These representations are processed by a world model (comprising a dynamics decoder and reward decoder). A Decision Transformer then conditions on this representation to predict future actions. In our experiments, as we focus exclusively on dynamics shifts, the reward decoder component is omitted.

**End-to-End Diffusion** (Lu et al., 2025). We adopt the modular architecture from the official implementation of DV (Lu et al., 2025), which decomposes the diffusion policy into three distinct components: a planner, reward guidance, and an optional inverse dynamics policy. In our deployment, we utilize a Diffusion Transformer (DiT) as the backbone for the planner. To inject dynamics information, we condition the planner on the history trajectory and utilize the diffusion model to inpaint the future trajectory. Consistent with DADP, we exclude the reward guidance module.

## B.3. The details of DADP

**Context Encoder Architecture**: The context encoder is implemented with a Transformer encoder with apative pooling at the output layer. Specifically, Two separate MLPs (state/ action encoder) map raw state and actions to the same dimension, and interleave the state tokens and action tokens to form a token sequence. Next, add learnable positional representations and apply dropout, and feed the position-augmented tokens into a Transformer encoder to produce output. Finally, aggregate the sequence using a learnable-query multi-head attention pooling to get a single vector as the representation.

**Diffuion Policy Architecture**: The diffusion policy in DADP is adapted from the official implementation of DV. We modify the forward and denoising processes according to the formulations in (9)–(14). All other components, including model hyperparameters and network architecture, remain identical to the original DV implementation.

The corresponding hyperparamters are shown in Table 5.

*Table 5.* Context Encoder Hyperparameters and Experimental Settings

| Hyperparameter | Value |
| --- | --- |
| *Context Encoder Architecture* | |
| Model Dimenstion | 256 |
| MLP Hidden Dimension | 256 |
| Feedforward Hidden Dimension | 1024 |
| Hidden Layer | 4 |
| Adaptive Pooling Heads | 8 |
| Adaptive Pooling Dropout | 0.1 |
| Attention Heads | 8 |
| History Length ($H$) | 16 |
| Task Representation Dim. | $\dim(s) + \dim(a)$ |
| *Context Encoder Training* | |
| Batch Size | 128 |
| $\beta_{\text{forward}}$ | 1.0 |
| $\beta_{\text{inverse}}$ | 1.0 |
| Training Ratio | 0.8 |
| Learning Rate | 3e-4 |
| Epochs | 10 |
| *Policy Architecture* | |
| Hidden Dimension | 256 |
| Planner Depth | 6 |
| Attention Heads | 8 |
| History Length ($H$) | 16 |
| Prediction Horizon | 4 |
| Noise Schedule | Cosine |
| *Policy Training* | |
| Batch Size | 256 |
| Learning Rate | 3e-4 |
| MuJoCo Iterations | 1e6 (Walker), 4e5 (Ant, Hopper), 1e5 (HalfCheetah) |
| Adroit Iterations | 5e5 (Relocation), 1e5 (Door) |
| *Inference & Evaluation* | |
| Inference Steps | 5 |
| Guidance Scale | 0.1 (Ant: 0.05) |
| Max Env Steps | 1000 (MuJoCo), 200 (Adroit) |
| Eval. Episodes | 50 (MuJoCo), 200 (Adroit) |

# C. Addtional Experiments

## C.1. Lagged Context Ablation on Meta-DT

To further validate that the lagged context idea is not specific to our diffusion-based pipeline, we apply the same $\Delta t$ ablation to Meta-DT (Wang et al., 2024b) (a Decision Transformer-based meta-RL baseline) across both domain-adaptive and reward-changing benchmarks. As shown in Table 6, increasing $\Delta t$ from 1 to 32 yields consistent gains on both domain-adaptive environments (Hopper-Param, Walker-Param) and the majority of reward-changing environments (Ant-Dir, Cheetah-Dir, Cheetah-Vel), with only a small regression on Point-Robot. This indicates that the static-domain disentanglement effect transfers across (i) different policy architectures and training pipelines and (ii) reward-variation settings, supporting the generality of the proposed representation-learning technique.

*Table 6.* Meta-DT performance with different $\Delta t$.

| Environment | $\Delta t = 1$ | $\Delta t = 32$ |
|---|---|---|
| **Point-Robot** | **-10.7** | $-11.6$ |
| **Ant-Dir** | 368.9 | **391.7** |
| **Cheetah-Dir** | 542.4 | **554.2** |
| **Cheetah-Vel** | $-100.1$ | **-98.7** |
| **Hopper-Param** | 342.0 | **363.6** |
| **Walker-Param** | 397.4 | **399.8** |

## C.2. Effect of different guidance scale

In this section, we conduct an ablation study over a range of values for the introduced guidance scale. As shown in Table 7, the results indicate that our method is not highly sensitive to this coefficient, with performance degrading sharply only when the guidance scale becomes excessively large. This demonstrates that our approach can achieve excellent performance without requiring extensive hyperparameter tuning.

*Table 7.* Ablation Results under Different Guidance Scales.

| Environment | Setting | 0 | 0.01 | 0.05 | 0.1 | 0.5 | 1 | 5 |
|---|---|---|---|---|---|---|---|---|
| **Walker2d** | Seen | 3722 | 4026 | **4231** | 3957 | 3968 | 3615 | 2759 |
| | Unseen | 2852 | 2721 | 2837 | 2681 | **2934** | 2891 | 1854 |
| **HalfCheetah** | Seen | 3509 | 3920 | 3808 | 4079 | **4114** | 4093 | 404.5 |
| | Unseen | 2496 | 2931 | 2604 | 3021 | 2934 | **3162** | 39.5 |

## C.3. Context Source

In DADP, since the representation represents static domain information, its context source is not restricted to online-collected recent history. In this section, we further consider several practical deployment settings to substantiate the properties of the learned representations and the general applicability of DADP. We continue to assume that, in an unknown domain, no ground-truth policy rollouts are available, as this represents the most realistic scenario when deploying a policy to a new domain. Specifically, we consider the following three variants of context source:

- **Cold Start**: DADP adopted approach, whose context is online collected recent history. When the online recent history length is insufficient, padding states and actions are used to complete the context window.

- **Persistent Context**: By executing the Cold Start policy, we have the in-domain policy rollouts as context source. We randomly sample a clip in the policy rollouts as the persistent context used during online inference.

- **Warm Start**: Following the Persistent Prompt, we replace the context source from policy rollouts to online recent history when length is sufficient. The context from policy rollouts is only used as warm start prompt.

*Table 8.* Ablation Results on Context Source. The normalized metric is the averaged performance normalized with the Expert Seen.

| Variants | Walker2d | | HalfCheetah | | Hopper | | Ant | | Normalized | |
|---|---|---|---|---|---|---|---|---|---|---|
| | Seen | Unseen | Seen | Unseen | Seen | Unseen | Seen | Unseen | **Seen** | **Unseen** |
| Cold Start | 3985 | 2765 | 4079 | 3045 | 1692 | 1809 | 3069 | 3414 | 0.851 | 0.797 |
| Persistent Context | 3988 | 2938 | 4080 | 2902 | 1679 | 1828 | 3206 | 3552 | 0.858 | **0.809** |
| Warm Start | 4117 | 2833 | 4070 | 2846 | 1688 | 1662 | 3221 | 3670 | **0.865** | 0.783 |

We evaluate the different variants with the same checkpoint across the MuJoCo environments. As shown in Table 8, different variants achieve comparable performance, further validating the static nature of the learned representations and the resulting flexibility in the choice of context sources. Moreover, this property enables the use of pre-collected rollouts obtained by executing the policy in the environment to mitigate the cold-start phase with incomplete context, thereby improving stability and performance once the context is fully populated.

## C.4. OOD (Out-of-Support) Parameter Ranges

This subsection documents the exact parameter ranges used for the **OOD** column in the main benchmark table (Table 1). For each MuJoCo environment, OOD parameters are sampled from intervals that lie *outside* the training factor space (specified in Table 4):

- **Walker2d**: Friction coefficient for the two feet $\in [1.07, 1.12] \cup [2.48, 2.52]$ (Training support: $[1.12, 2.48]$).

- **Hopper**: Joint damping for three joints and friction $\in [0.65, 0.75] \cup [1.19, 1.30]$ (Training support: $[0.75, 1.19]$).

- **HalfCheetah**: Torso length $\in [0.20, 0.24] \cup [1.05, 1.11]$ (Training support: $[0.336, 0.951]$).

- **Ant**: Length of four legs $\in [0.34, 0.43] \cup [1.65, 1.90]$ (Training support: $[0.465, 1.612]$).

For each environment, we evaluate five randomly-sampled out-of-support parameter sets. The numerical results are reported in the OOD rows of Table 1: DADP maintains performant behavior and exhibits clear advantages over all baselines under genuine out-of-support extrapolation across all four MuJoCo environments.

## C.5. Non-stationary Dynamics Evaluation

DADP is designed for stationary settings where the domain parameters remain constant within an episode. Nonetheless, because the online context is always drawn from the most recent history, the encoder can in principle track slowly varying or piecewise-stationary dynamics by re-estimating the representation from up-to-date context. We empirically validate this by deploying the *same* Walker2d checkpoint (trained under stationary dynamics) on four non-stationary friction schedules:

- **Increasing**: friction coefficient uniformly increasing in $[1.32, 2.32]$ over the episode.

- **Decreasing**: friction coefficient uniformly decreasing in $[2.28, 1.28]$ over the episode.

- **Random**: friction coefficient resampled from $[1.32, 2.28]$ every 50 steps.

- **Leaping**: friction coefficient alternates between $\{1.32, 2.28\}$ every 50 steps.

*Table 9.* DADP performance on Walker2d under non-stationary friction schedules. The same checkpoint is used across all settings; "Seen" reports the original stationary Seen performance for reference.

| Mode | Seen | Increasing | Decreasing | Random | Leaping |
|---|---|---|---|---|---|
| DADP | $4100 \pm 85$ | $4348 \pm 144$ | $4105 \pm 251$ | $4194 \pm 129$ | $3772 \pm 128$ |

As shown in Table 9, DADP remains performant across all four non-stationary schedules, confirming that the up-to-date online context allows the static encoder to track piecewise- or slowly-changing dynamics in practice.

## C.6. Inference Efficiency: One-Step Generation

In our main experiments, DADP uses only 5 DDIM inference steps (Table 5), introducing no extra cost per denoising step relative to a standard diffusion policy. Beyond this, DADP's representation-biased prior shifts the diffusion start point closer to the target action manifold, which we hypothesize makes the generation more amenable to aggressive step reduction. We test this by comparing DADP and a standard end-to-end diffusion policy under both 5-step and 1-step DDIM sampling on the two locomotion environments. Performance numbers in parentheses indicate the percentage performance drop versus the 5-step variant of the same method.

*Table 10.* Inference-efficiency comparison: 5-step vs 1-step DDIM sampling. The values in parentheses indicate the percentage performance drop relative to the 5-step variant of the same method.

| Environment | Diffusion (5-step) | DADP (5-step) | Diffusion (1-step) | DADP (1-step) |
|---|---|---|---|---|
| Walker2d | 3722 | **3991** | 158 ($\downarrow$ 85.8%) | **2830** ($\downarrow$ 29.1%) |
| HalfCheetah | 3509 | **4100** | 504 ($\downarrow$ 85.6%) | **1357** ($\downarrow$ 66.9%) |

As shown in Table 10, while a standard diffusion policy collapses under one-step inference (losing $\sim$86% performance), DADP retains a substantial fraction of its performance, providing a clear practical advantage for compute-constrained real-time control deployments.

## C.7. Per-Environment Notes and Extended Utilization Ablation

**Extended utilization ablation.** We additionally report the representation-utilization ablation on Hopper and Ant, mirroring Table 3 in the main paper. The qualitative trend is consistent: DADP's diffusion-injection utilization remains the strongest variant overall, confirming that the utilization findings generalize across all four MuJoCo locomotion environments.

*Table 11.* Representation-utilization ablation on Hopper and Ant under the same variants as Table 3.

| Variants | Hopper Seen | Hopper Unseen | Ant Seen | Ant Unseen |
|---|---|---|---|---|
| End-to-End Diffusion | 1634 | 1701 | 2955 | 3394 |
| Conditional Policy | 1265 | 1345 | 2527 | 2764 |
| Better Representation | **1710** | **1760** | 2650 | 3229 |
| Mixed DDIM | 1629 | 1687 | 3031 | **3527** |
| **DADP (Ours)** | 1643 | 1711 | **3117** | 3495 |

## C.8. Direct Entanglement Diagnostic: Within-Domain Std of $z_t$

To provide direct diagnostic evidence that larger $\Delta t$ removes time-varying components from the learned representation, we report the within-domain standard deviation of $z_t$ (averaged across domains, normalized by the supervised-encoder reference) as a function of $\Delta t$. A representation that captures only static information should remain essentially constant within a single domain, yielding a low in-domain std.; entanglement with time-varying signals would increase this value.

*Table 12.* Within-domain standard deviation of $z_t$ (normalized w.r.t. supervised reference). Lower values indicate better disentanglement of static domain information from transient signals.

| Environment | Metric | $\Delta t = 1$ | $\Delta t = 4$ | $\Delta t = 16$ | $\Delta t = 32$ | $\Delta t = \infty$ | Supervised |
|---|---|---|---|---|---|---|---|
| Walker2d | In-domain Std. | 14.2 | 8.2 | 7.8 | 7.3 | **0.9** | 1.0 |
| HalfCheetah | In-domain Std. | 8.9 | 6.8 | 4.7 | 1.7 | **0.9** | 1.0 |

As shown in Table 12, the in-domain std monotonically decreases as $\Delta t$ grows, and at $\Delta t = \infty$ the value matches the supervised reference. This is direct evidence — complementary to the linear-probe accuracy and reconstruction-loss results in Table 2 — that the lagged context mechanism progressively disentangles static domain information from transient dynamical signals.

# D. Visualization

## D.1. Representation Visualization

We also provide the t-SNE visualizations of the representations of HalfCheetah learned with different $\Delta t$ in figure 6. As $\Delta t$ increases, the representations from different domains also become gradually distinctly clustered.

We observe from the quantitative results that, in Walker2d, increasing $\Delta t$ yields a substantially larger improvement in representation quality compared to HalfCheetah. This trend is also reflected in the visualization: when $\Delta t = 1$, different domains already exhibit partial clustering behavior, and for some domains, increasing $\Delta t$ leads to improved cluster separation. As $\Delta t \to \infty$, the resulting representations achieve high quality comparable to those observed in the Walker2d setting.

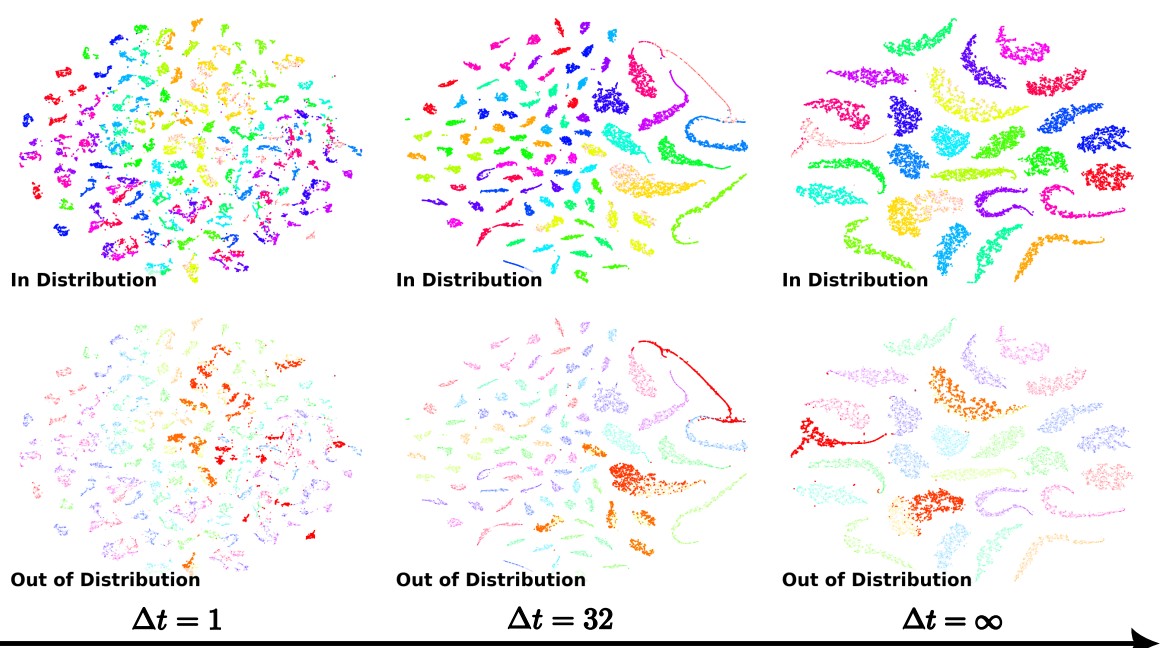

*Figure 6.* t-SNE visualization of half cheetah representations learned with different $\Delta t$.

## D.2. Domain-specific Action Modalities

In this section, we provide visualizations of different domain-specific gaits presented in different tasks in Figure 7, 8, 9, 10.

As mentioned in Appendix B.1, we do not introduce morphological variations in the Hopper environment for better and more stable expert data generation. As shown in Figure 7, without morphological variations, the gaits across different domains are similar, resulting in reduced data diversity. This aligns with our analysis on previous benchmarks.

As shown in Figure 8, 9, 10, it is clear that by introducing morphological variations during the data generation phase, the gaits and action modalities across different domains become substantially more diverse, thereby constructing a more challenging domain adaptation benchmark. Despite that, our proposed method is able to achieve state-of-the-art performance in environments with substantial dynamical gaps, demonstrating its broader applicability compared to prior approaches.

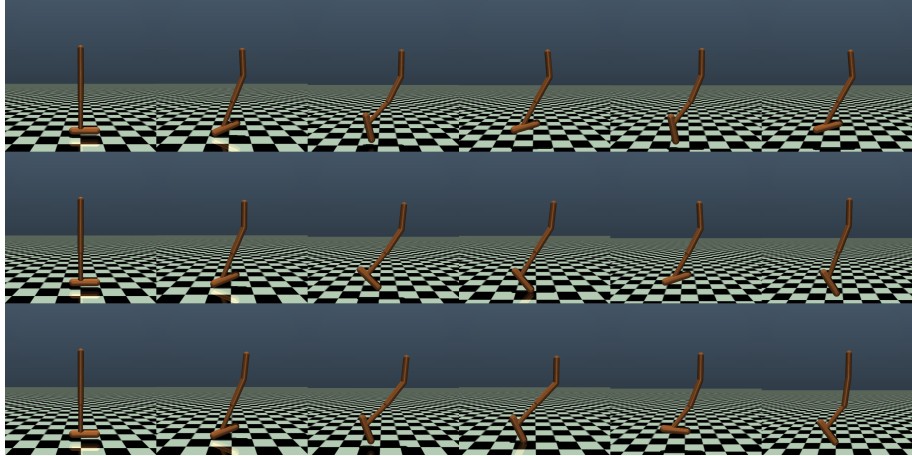

*Figure 7.* Different Domain-specific Gaits in Hopper. Without morphological variations, the gaits are similar across different domains.

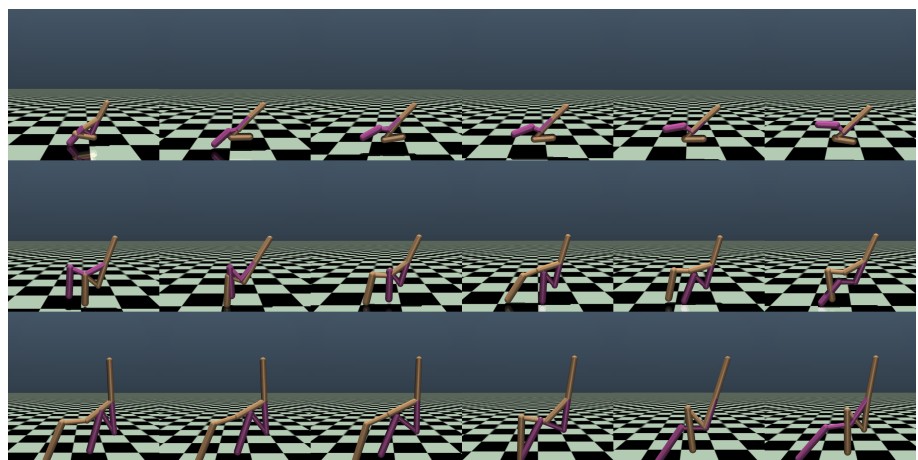

*Figure 8.* Different Domain-specific Gaits in Walker

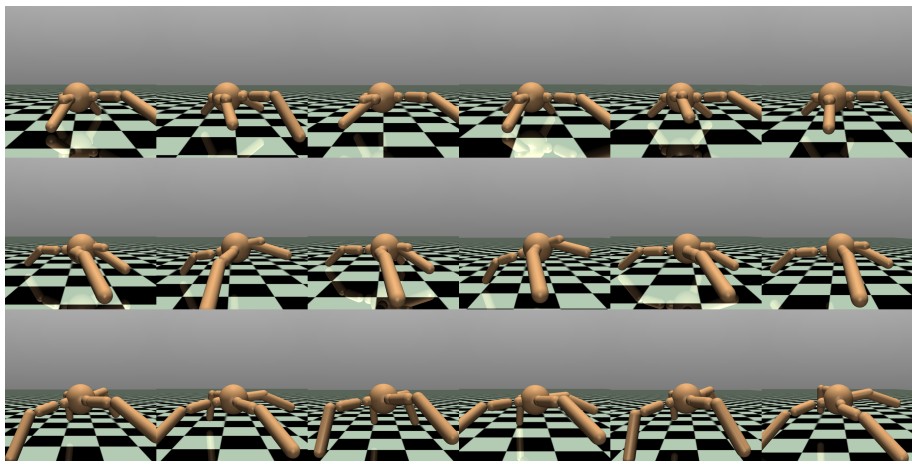

*Figure 9.* Different Domain-specific Gaits in Ant

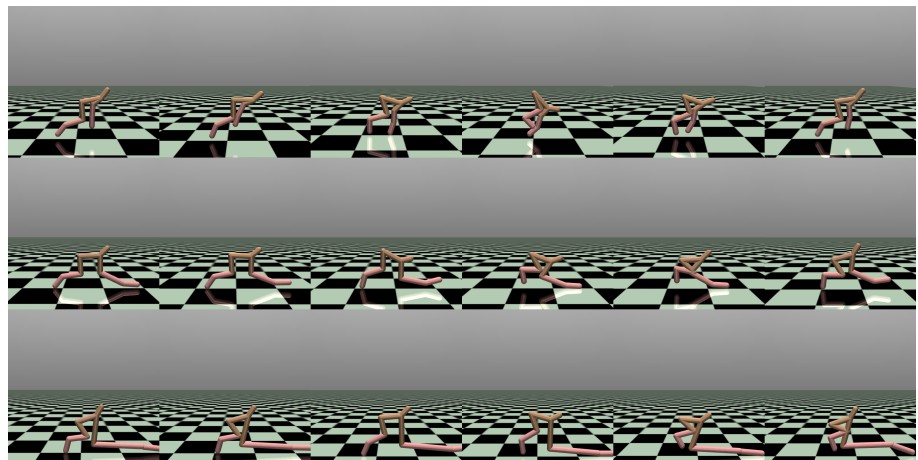

*Figure 10.* Different Domain-specific Gaits in HalfCheetah

## D.3. Mastery Level

In this section, we provide visualizations of the mastery. Here, mastery refers to a policy's ability to successfully handle multiple domains, reflecting whether a single policy can robustly control different domains or embodiments, which directly impacts its practical effectiveness. As show in Figure 11, in Walker2d, a policy with high mastery level is able to run forward rapidly and stably for an extended duration (top row). In contrast, policies that achieve non-zero returns but fail to reach mastery exhibit suboptimal gaits (middle row), or even collapse and fall (bottom row). A similar pattern can be observed in HalfCheetah, as illustrated in Figure 12.

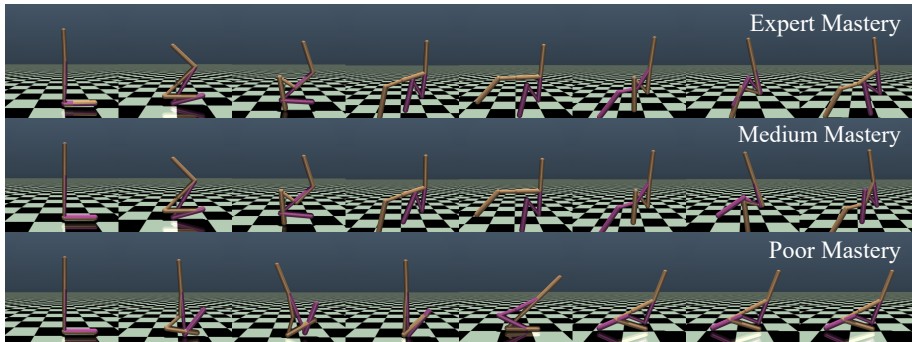

*Figure 11.* Mastery Level Visualization in Walker2d Environments

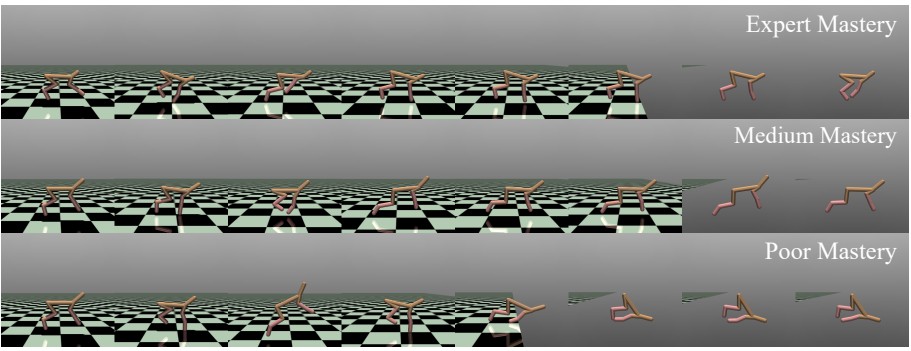

*Figure 12.* Mastery Level Visualization in HalfCheetah Environments

## E. Pesudocodes

In this section, we present the pesudocodes of the proposed DADP pipeline.

---

**Algorithm 1** Domain Adaptive Diffusion Policy (DADP)

---

**Domain Representation Learning**

**Input**: Offline datasets $\mathcal{D}$, context encoder $E_\phi$, forward dynamics model $f_\theta$, inverse dynamics model $f_\psi$
**for** each epoch **do**
  Rebuild random domain-based paired index $j = f(i)$
  **while** epoch not finished $\mathcal{D}$ **do**
    Sample paired data $\tau^i, s^j_{t_j}, a^j_{t_j}, s^j_{t_j+1}$
    Get context representation $z^i = E_\phi(\tau^i)$
    Get dynamical prediction $\hat{s}^j_{t_j+1} = f_\theta(s^j_{t_j}, a^j_{t_j}, z^i)$, $\hat{a}^j_{t_j} = f_\psi(s^j_{t_j}, s^j_{t_j+1}, z^i)$          // *Cross-Prediction*
    Update $\phi, \theta, \psi$ with loss: $\mathcal{L}(\phi, \theta, \psi) := \beta_{\text{forward}} \left\| \hat{s}^j_{t_j+1} - s^j_{t_j+1} \right\|^2 + \beta_{\text{inverse}} \left\| \hat{a}^j_{t_j} - a^j_{t_j} \right\|^2$
  **end while**
**end for**

---

**DADP Training**

**Input:** Dataset $\mathcal{D}$, Denoising Schedule $\alpha_{1:K}, \sigma_{1:K}$, Guidance Scale $\lambda$, Denoising Policy $\epsilon_\xi$
**while** not converged **do**
  **1. Data Sampling & Preparation**
    Sample batch $(\tau, z_{\text{pre}}) \sim \mathcal{D}$          // *Sample trajectories and precomputed representations*
    Set $x_0 = \tau$ and $M$ s.t. $M_{t<H} = 1$ and $M_{H,\text{obs}} = 1$          // *Initialize clean data and history mask*
  **2. Forward Process**
    Sample time $k \sim \text{Uniform}(0, 1)$, noise $\epsilon \sim \mathcal{N}(0, I)$
    $x_k^{\text{raw}} = \alpha_k(x_0 - \lambda z) + \lambda z + \sigma_k \epsilon$          // *Mixed guassian prior*
    $x_k = (1 - M) \odot x_k^{\text{raw}} + M \odot x_0$          // *Keep History for inpainting*
  **3. Reverse Process**
    $\hat{\epsilon} = \epsilon_\xi(x_k, k)$
    Update with Loss: $\mathcal{L}_\xi = \|(\sigma_k \epsilon + (1 - \alpha_k)\lambda z - \hat{\epsilon}) \odot (1 - M)\|^2$
**end while**

---

**DADP Evaluation**

**Input:** History Context $h_{\text{ctx}}$, Sampling Steps $S$, Guidance Scale $\lambda$, Context Encoder $E_\phi$, Denoising Policy $\epsilon_\xi$
**Output:** Action $a_H$
    Compute Representation $z = E_\phi(h_{\text{ctx}})$
    Sample $\epsilon \sim \mathcal{N}(0, I)$,
    Set $M$ s.t. $M_{t<H} = 1$ and $M_{H,\text{obs}} = 1$          // *Initialize with mixed noise and masking*
    $x_K^{\text{raw}} = \lambda z + \epsilon$          // *Sample from mixed guassian*
    $x_K = (1 - M) \odot x_K^{\text{raw}} + M \odot h_{\text{ctx}}$          // *Keep History for inpainting*
  **2. Denoising Loop**
  **for** $i = S, \ldots, 1$ **do**
    Map $i$ to diffusion times $k$ and $k - 1$
    $\hat{\epsilon} = \epsilon_\xi(x_k, k)$          // *Noise Prediction*
    $x_{k-1}^{\text{raw}} = \frac{\alpha_{k-1}}{\alpha_k}(x_k - \hat{\epsilon}) + \frac{\sigma_{k-1}}{\sigma_k}\hat{\epsilon}$          // *Denoise Step*
    $x_{k-1} = (1 - M) \odot x_{k-1}^{\text{raw}} + M \odot h_{\text{ctx}}$          // *Keep History for inpainting*
  **end for**
  **3. Execution**
    Extract action: $a_H = x_0[H, \text{actions}]$
    **return** $a_H$

---

