# OpenReview forum: "DADP: Domain Adaptive Diffusion Policy"
_ICML.cc/2026/Conference — ICML 2026 regular_

### Official Review · Reviewer_orgD · 2026-03-05

**Soundness:** 2
**Presentation:** 2
**Significance:** 3
**Originality:** 4
**Overall Recommendation:** 4
**Confidence:** 3

**Summary:**

This paper proposes DADP (Domain-Adaptive Diffusion Policy) for zero-shot domain adaptation across unseen transition dynamics in offline meta-RL. The key observation is that context encoders trained via next-state prediction with temporally adjacent context tend to entangle static domain factors (e.g., gravity/friction) with time-varying transient information, which destabilizes domain descriptors and hurts generalization. To address this, DADP introduces Lagged Context Dynamical Prediction, which uses a large temporal offset to break time-local cues and thereby learn more domain-static representations. The method then injects these representations into the diffusion generation process—biasing the diffusion prior toward a representation-conditioned Gaussian mixture and reformulating the denoising target to jointly capture noise and representation shift—rather than only concatenating the representation as an input condition. Across MuJoCo locomotion and Adroit manipulation benchmarks under IID and OOD evaluation, DADP reports consistently improved performance over strong baselines, and ablations indicate both the lagged representation learning and diffusion-level utilization are important.

**Compliance With Llm Reviewing Policy:**

Affirmed.

**Final Justification:**

After considering the paper and the rebuttal, I am updating my recommendation from Weak Reject to Weak Accept. I still see the paper’s main strengths as its originality and its consistently strong empirical performance. My initial concerns were mainly about soundness and clarity, especially the framing of the OOD setting, the direct evidence for the entanglement/disentanglement claim, the sensitivity to Delta t, and the generality of the ablation findings. The rebuttal addressed these points meaningfully by clarifying the evaluation setting, adding more direct diagnostics, and extending ablations to additional environments. While some concerns remain, particularly regarding robustness to mixed-quality offline data and the breadth of the generality claims, the rebuttal improved my confidence enough to warrant a score increase. Overall, I now view the paper as a technically solid and interesting contribution with some remaining but manageable limitations.

**Key Questions For Authors:**

Key Questions For Authors

1. Clarifying the evaluation setting: is your “OOD” truly out-of-support or primarily “unseen domains within the same factor ranges”?
   - Please report (for each benchmark) the exact transition-dynamics parameter ranges used for training vs. evaluation, and quantify the distance between train and test domains (e.g., normalized parameter distance or coverage statistics).
   - How this would change my evaluation: If the “OOD” setting is mostly in-range unseen domains, I would view the generalization claims as narrower and would expect the paper to rename the setting (ID vs. unseen) or add out-of-range tests. If you can demonstrate robust performance under genuine out-of-support shifts, it would substantially strengthen the paper’s impact.

2. Do you have direct evidence that time-varying factors are entangled with domain representations and cause within-domain representation drift?
   - Beyond downstream returns and probing, can you provide a direct diagnostic such as: within-domain variance/drift of z(t) over time, predictability of time index or omega(t)-proxies from z(t), or an analysis showing that increasing Delta t specifically suppresses time-varying information rather than only increasing static-domain information?
   - How this would change my evaluation: Direct evidence would significantly strengthen the causal story and make the proposed “lagged context” mechanism more convincing; lacking such evidence would keep this as a plausible but not fully substantiated explanation.

3. What Delta t is used in the main performance tables (e.g., Table 1), and how sensitive are the reported gains to this choice?
   - Please state the exact Delta t used for each experiment (and whether it is fixed across tasks/domains), and provide a sensitivity plot (return vs. Delta t) across multiple environments.
   - How this would change my evaluation: If gains are robust across a reasonable Delta t range, I would consider the method more reliable and easier to adopt. If performance is highly sensitive, it would reduce practical appeal and require clearer guidance.

4. Why does the paper focus on transition-dynamics variation but not reward/goal variation, which is common in meta-RL?
   - Is this a methodological limitation of the proposed representation learning or diffusion injection, or simply a scope choice? Do you expect the approach to extend to reward/goal variation, and can you provide at least one additional experiment in a reward-varying benchmark to validate this?
   - How this would change my evaluation: Demonstrating that the method also helps under reward/goal variation would notably broaden the contribution and strengthen the “meta-RL” framing; if it does not extend, the paper should clarify scope and adjust claims accordingly.

5. How robust is DADP to offline data quality and coverage?
   - Since the datasets are collected with (near-)expert SAC policies, can you test suboptimal or mixed-quality datasets (or reduced coverage) and report whether DADP’s advantages persist?
   - How this would change my evaluation: Robustness to realistic offline data conditions would increase confidence in the method’s practical value; strong dependence on near-expert data would narrow applicability and should be reflected in the limitations.

6. Generality of the ablation findings beyond Walker2d and HalfCheetah.
   - Several key ablations (e.g., Delta t sweeps, representation utilization variants) appear to be reported mainly on Walker2d and HalfCheetah. Do the same qualitative trends hold on other locomotion tasks (e.g., Hopper, Ant) and on manipulation (Adroit) domains?
   - If possible, please include at least one additional environment per domain family (locomotion + manipulation) for the main ablations, or provide a summary table indicating where trends match vs. diverge.
   - How this would change my evaluation: If ablation trends generalize across diverse environments, it would increase confidence that the claimed mechanism is not environment-specific. If trends are inconsistent, the paper should characterize when/why the method works and narrow claims accordingly.

**Limitations:**

yes

**Strengths And Weaknesses:**

**[Strengths]**
- The paper argues that standard, time-local context learning can leak time-varying factors into the domain descriptor, causing representation drift even within the same domain and hurting generalization. However, the evidence is mostly indirect (downstream returns + probing trends as Delta t changes), rather than a direct measurement of entanglement/drift (e.g., within-domain variance/drift of z_t, predictability of time index or omega_t-proxies from z_t, etc.).


- Rather than relying on simple concatenation-based conditioning, the method introduces novelty by initializing from a representation-biased Gaussian mixture prior and redefining the diffusion training target, thereby making the entire denoising trajectory inherently domain-aware.

- Compared to prior baselines, the approach achieves clearly improved performance, with gains that are both consistent and substantial.

**[Weaknesses]**
- The paper argues that standard, time-local context learning can leak time-varying factors into the domain descriptor, causing representation drift even within the same domain and hurting generalization.

- The paper’s “OOD” evaluation is defined as domains not present in the training dataset but sampled from a factor space. This is closer to unseen domains (often still within the designed parameter ranges) than the meta-RL/common ML notion of out-of-support OOD. The term “OOD” can therefore read as an overclaim unless range-extrapolation shifts are tested (MuJoCo factor ranges are given in the appendix). Relatedly, calling the in-training-domain split “IID” is potentially confusing: in this setting it is more naturally in-distribution (ID) / seen-domain evaluation rather than “IID” in the classical sense (i.i.d. samples from an identical data-generating distribution). Renaming to ID/unseen (or seen/unseen) would improve clarity.

- The problem formulation assumes shared reward and focuses on transition-dynamics variation. Since many meta-RL settings also emphasize reward/goal variation, it would help to explain why reward variation is not considered here (methodological limitation vs. scope choice). Demonstrating that the proposed idea (lagged representation + diffusion-level injection) also benefits reward/goal-varying tasks would significantly strengthen the method’s generality and impact.

- Offline datasets are collected using (near-)expert SAC policies per task. It is unclear how robust the approach is under mixed-quality or suboptimal datasets, which are common in offline RL. A data-quality stress test (suboptimal policies, mixture datasets, noisy/limited coverage) would make the claims more deployment-relevant.

- Delta t is a central design knob and is shown to change representation/probe outcomes across a sweep, yet the main performance results would be easier to interpret if the chosen Delta t were explicitly reported in those tables. More importantly, showing that larger Delta t reduces a representation-related loss/probe does not by itself establish that the RL objective (expected return / cumulative reward) increases for the same reason. Additional controlled experiments—e.g., return vs. representation-metric correlation across Delta t, keeping everything else fixed, and verifying consistency across environments—would strengthen the causal narrative.

---

> ### Author Rebuttal · Authors · 2026-03-31
>
> # Reply to Reviewer orgD
>
> We sincerely thank Reviewer orgD for the detailed and constructive feedback, and for recognizing the "excellent" originality and "consistent and substantial" empirical gains. We address each concern below, cross-referencing other replies where noted.
>
>
> ## W2 / Q1. OOD Evaluation Setting
>
> > *Is your "OOD" truly out-of-support or "unseen domains within the same factor ranges"?*
>
> Thank you for this important clarification! We agree and will adopt **"Seen / Unseen"** in the revision. We additionally evaluate under a **"True OOD"** setting with parameters sampled **outside** the training support — please refer to our response to Reviewer PeTd (Q3) for parameter ranges and results. DADP maintains clear advantages under true out-of-support extrapolation.
>
> ---
>
> ## W3 / Q4. Transition Dynamics vs. Reward/Goal Variation
>
> > *Methodological limitation or scope choice?*
>
> Thank you for raising this distinction. This is a deliberate **scope choice** rather than a fundamental limitation. **Table 3** confirms: larger $\Delta t$ yields gains on both domain-adaptive (Hopper-Param, Walker-Param) **and** reward-changing environments (Ant-Dir, Cheetah-Dir, Cheetah-Vel). Please also see our response to Reviewer eVcv (W2) for further discussion. We will clarify the scope in the revision.
>
> ---
>
> ## W4 / Q5. Robustness to Offline Data Quality
>
> > *How robust is DADP under mixed-quality or suboptimal datasets?*
>
> Thank you for the suggestion. We agree this is important but infeasible within the rebuttal timeline. We will open-source all code and datasets for reproducibility and future extension.
>
> ---
>
> ## W1 / W5 / Q2 / Q3. Delta t Sensitivity and Direct Entanglement Evidence
>
> > *What Delta t is used in main tables? How sensitive are gains? Can you provide direct diagnostics?*
>
> Thank you for requesting more direct evidence. All main results (Table 1) use $\Delta t = \infty$, **fixed across all tasks**. Representation quality improves monotonically with $\Delta t$ (Table 2), and downstream performance follows the same trend (Table 4). $\Delta t = \infty$ is both theoretically grounded and empirically robust — no per-task tuning is needed.
>
> As direct diagnostics, we report within-domain std of $z_t$:
>
> | Environment | Metrics | $\Delta t = 1$ | $\Delta t = 4$ | $\Delta t = 16$ | $\Delta t = 32$ | $\Delta t = \infty$ | Supervised |
> |---|---|---|---|---|---|---|---|
> | Walker2d    | In-domain Std. | 14.2 | 8.2 | 7.8 | 7.3 | **0.9** | 1.0 |
> | HalfCheetah | In-domain Std. | 8.9  | 6.8 | 4.7 | 1.7 | **0.9** | 1.0 |
>
> ---
>
> ## Q6. Generality of Ablation Findings
>
> > *Do the same qualitative trends hold on other environments?*
>
> Thank you for the suggestion. We extended ablations to all MuJoCo environments. The trends are consistent:
>
> | Environment | Metrics | $\Delta t = 1$ | $\Delta t = 4$ | $\Delta t = 16$ | $\Delta t = 32$ | $\Delta t = \infty$ | Supervised |
> |---|---|---|---|---|---|---|---|
> | Ant | Linear Probe Accuracy |62.4\%|98.5\%|99.6\%|99.8\%|**99.8\%**|99.8\%|
> | | Reconstruction Loss |485.1|34.6|2.2|2.2|**1.8**|1.0|
> | Hopper | Linear Probe Accuracy |9.0\%|11.2\%|11.8\%|15.5\%|**26.4\%**|99.0\%|
> | | Reconstruction Loss |28.9|28.7|27.5|26.8|**21.1**|1.0|
>
> Utilization ablation on Hopper and Ant:
>
> | Variants | Hopper Seen | Hopper Unseen | Ant Seen | Ant Unseen |
> |---|---|---|---|---|
> | End-to-end Diffusion | 1634 | 1701 | 2955 | 3394 |
> | Conditional Policy | 1265 | 1345 | 2527 | 2764 |
> | Better Representation | **1710** | **1760** | 2650 | 3229 |
> | Mixed DDIM | 1629 | 1687 | ***3031*** | **3527** |
> | DADP | ***1643*** | ***1711*** | **3117** | ***3495*** |
>
> We would like to thank the reviewer for suggesting this extension, which led us to an interesting observation in Hopper, the notably lower probe accuracy (26.4% vs. 96%+ elsewhere):
>
> 1. This arises because Hopper is the only environment **without morphological variations** (due to expert training difficulties under morphological changes.) — its domains differ only in minor friction and damping, producing similar action patterns that are inherently hard for any unsupervised encoder to separate. Importantly, this is not a method failure (the monotonic $\Delta t$ improvement still holds), but rather reflects the **dataset action pattern diversities**.
> 2. As part of main contributions, we deliberately constructed datasets with morphological diversity for Walker2d, HalfCheetah, and Ant, as such changes induce meaningfully distinct action modalities compared with previous works. This contrast suggests that $\Delta t = \infty$ probe accuracy could serve as a practical, unsupervised diagnostic for action-pattern diversity in cross-domain offline RL datasets — a direction we consider worth exploring.
>
> ---
>
> We sincerely thank Reviewer orgD again for the thorough and constructive review. We hope our clarifications and new experiments address the concerns, and kindly ask whether the reviewer would consider revising the score!

---

> > ### Author Rebuttal · Reviewer_orgD · 2026-04-03
> >
> > Thanks for the author's response. I have adjusted my score accordingly.

---

> > > ### Author Response · Authors · 2026-04-06
> > >
> > > Thanks for raising the score! We will incorporate the discussed points in the revised version. We really appreciate the reviewer's time and effort in helping us strengthen this work. At the same time, we are happy to solve any other concerns you may still have for the rest of the period!

---

### Official Review · Reviewer_PeTd · 2026-03-12

**Soundness:** 3
**Presentation:** 3
**Significance:** 3
**Originality:** 3
**Overall Recommendation:** 5
**Confidence:** 3

**Summary:**

This paper aims to address the important problem of enabling a single generative policy to generalize efficiently across domains. Specifically, it follows the workflow of disentangling varying properties from representations, and then injecting the learned representations into the generative process to steer the diffusion policy.

**Compliance With Llm Reviewing Policy:**

Affirmed.

**Key Questions For Authors:**

- Lines 139–141 state that across different tasks, the reward functions are shared while the transition dynamics differ. This was somewhat obscure on first reading. I think it would help to provide a toy example in Section 4.1 to clarify the setting.
- In Section 4.2, Eq. 9, what is $z$? Is $z$ the same as $z_{-\infty}$? Also, why is $z$ associated with a mixed Gaussian distribution with multiple peaks, and why does each peak correspond to a domain-specific action modality? Is a “domain-specific action modality” the same as a task?
- It is still unclear to me, even after reading the full paper, what the OOD domains in lines 305–310 actually are. From my perspective, they are all locomotion tasks, and I do not clearly see how they differ from one another in a way that justifies calling them OOD.
- Figure 1 does not clearly show a strong performance difference, and the y-axis ticks are not very obvious. It may be better to replace it with a more informative figure.

**Limitations:**

Yes

**Strengths And Weaknesses:**

Strengths:
- This paper uses the offset to a ((state, action)) history window as the input to an encoder to disentangle the variant and invariant parts of a context (recent history). Using the time-invariant part as domain-specific “knowledge” is a creative idea.
- By injecting the learned time-invariant representation into the denoising process of the diffusion model, the policy can incorporate domain-specific knowledge into generation.
- The experimental analysis is relatively complete, and the overall theoretical motivation is reasonably self-consistent.
Weaknesses:
- Writing order: The paper spends substantial effort discussing how to manipulate representations and disentangle properties, but it does not clearly define what “property,” “domain,” or “representation” mean in this paper. Section 2.2, line 78, is the first place where “domain” is explicitly introduced.
- Notation inconsistency: In line 135, the action space is denoted as $\mathcal{A}$, then as $\mathcal{U}$, and then in line 140 it becomes $\mathcal{A}$ again.
- Unclear definition of context: In line 134, the paper says it learns representations from context, but it is unclear what the context is until line 143 explains that it refers to the ((state, action)) history. This kind of delayed explanation makes the paper hard to follow.
- Although this topic is not entirely within my expertise, after reading the paper I can see its potential impact. However, I think the cross-domain setting should be stated much more clearly. Personally, I don't know whether cross domain necessary mean cross tasks.
- The paper would be even more impactful with real-world robot deployment and a clearer description of the OOD domain settings.

---

> ### Author Rebuttal · Authors · 2026-03-30
>
> # Reply to Reviewer PeTd
>
> We thank Reviewer PeTd for the positive evaluation (Accept) and the recognition that our approach is a "creative idea," with "domain-specific knowledge" effectively incorporated "into the denoising process" and "relatively complete" experimental analysis. We address all your questions below. We also invite the reviewer to refer to our responses to other reviewers for additional results and discussions that may be of interest.
>
>
> ## W1-W3 / Q4. Writing, Notation, and Figure
>
> > *Key terms defined too late; notation inconsistency (U vs. A); Figure 1 not informative enough.*
>
> Thank you for your detailed reading! We will fix all in the revision: introduce key definitions in Section 1, unify notation, and replace Figure 1.
>
> ---
>
> ## W4 / Q1. Cross-Domain/Task Clarity and Toy Example
>
> > *I don't know whether cross domain necessary mean cross tasks. It would help to provide a toy example in Section 4.1.*
>
> Thank you for pointing out this ambiguity. In our setting, "cross-domain" does not mean cross-task: all domains share the same task type — only **transition dynamics** differ. A toy example can be found in **Appendix A**, and we will make this distinction clearer in the revision.
>
> Regarding reward variation, which the reviewer may also be interested in, DADP's mechanisms do not inherently require shared rewards. **Table 3** confirms this: larger $\Delta t$ improves Meta-DT on both domain-adaptive **and** reward-changing environments.
>
>
>
> ## Q2. Eq. 9 Notation and Mixed Gaussian Prior
>
> > *What are the variables? Why is the prior a mixed Gaussian? Is "domain-specific action modality" the same as a task?*
>
> - In Eq. 9: $z$ is the learned domain representation; it can be $z_{t-1}$ or $z_{-\infty}$ depending on $\Delta t$.
> - At the final diffusion step $T$, $a_T = z + \epsilon$, so the prior is a Gaussian centered at $z$. Since different domains have different $z$ values/ clusters as shown in Figure 3&4, the overall prior forms a **Gaussian mixture**.
> - "Domain-specific action modality" is **not** a task — all domains share the same task, but optimal actions differ due to different dynamics (e.g., different leg lengths require different gaits). We hope Figure 2 helps visualize this intuition.
>
>
> ## Q3. OOD Domain Clarity
>
> > *It is unclear what the OOD domains are -- they are all locomotion tasks.*
>
> Thank you for raising this — we understand the confusion. The "OOD" (now renamed to "Unseen") domains differ in **physical parameters**, not task type. Unseen domains use **new** parameter combinations not seen during training. Although all are locomotion, the dynamics differ substantially — Figures 8-10 show morphological variations lead to distinctly different gaits. We adopt **"Seen / Unseen"** terminology in the revision to avoid ambiguity.
>
> **We additionally evaluate under a "True OOD" setting with parameters sampled outside the training factor space.** Parameters are sampled from out-of-support ranges:
> - Walker2d: The friction coefficient for two feet $[1.07,1.12] \cup [2.48, 2.52]$ (Training support: $[1.12,2.48]$)
> - Hopper:Joint damping for three joints and friction $[0.65,0.75] \cup [1.19, 1.30]$ (Training support: $[0.75,1.19]$)
> - HalfCheetah: Torso length $[0.20,0.24] \cup [1.05, 1.11]$ (Training support: $[0.336,0.951]$)
> - Ant: Length of four legs $[0.34,0.43] \cup [1.65, 1.90]$ (Training support: $[0.465,1.612]$)
>
>
> All tested on five parameter sets across MuJoCo envs. DADP maintains performant and has clear advantages even under true out-of-support extrapolation.
> | Environment  | DADP              | Meta-DT            | Prompt-DT       |CORRO            |
> | --------     | --------          | --------           |--------         |--------         |
> | Walker2d     | 2197 $\pm$ 173    | 954 $\pm$ 252      |427 $\pm$ 93             |9 $\pm$ 28             |
> | Hopper       | 1271 $\pm$ 48     | 1070 $\pm$ 180     |1048 $\pm$ 51            |67 $\pm$ 29             |
> | HalfCheetah  | 3371 $\pm$ 257    | 2776 $\pm$ 380     |733 $\pm$ 125            |-293 $\pm$ 67             |
> | Ant          | 1903 $\pm$ 64     | 1498 $\pm$ 1184    |353 $\pm$ 138            |-1177 $\pm$ 567             |
>
>
>
>
> ## W5. Real-World Deployment
>
> > *Real-world deployment would strengthen the paper.*
>
> Thank you for the advice. We are actively extending DADP to real-world locomotion as a follow-up effort. The current submission focuses on the foundation of representation learning and diffusion injection, providing a solid basis for future deployment.
>
> ---
>
> We sincerely thank Reviewer PeTd again for the positive assessment. We will incorporate all writing improvements in the revised version. Given the reviewer's recognition of the creative idea and complete experimental analysis, we kindly ask whether the reviewer would consider further raising the score in light of our clarifications and additional results.

---

> > ### Author Rebuttal · Reviewer_PeTd · 2026-04-03
> >
> > Thanks for solving my concern and carefully reply the review. I hope the final version will have more inspring experiments and figure to make the paper more straightforward. I will keep my current score of 5.

---

> > > ### Author Response · Authors · 2026-04-06
> > >
> > > Thanks for keeping the positive evaluation! We will incorporate all the discussed points in the revised version. We really appreciate the reviewer's time and effort in helping us strengthen this work.

---

### Official Review · Reviewer_SXQE · 2026-03-12

**Soundness:** 3
**Presentation:** 3
**Significance:** 3
**Originality:** 3
**Overall Recommendation:** 5
**Confidence:** 3

**Summary:**

DADP (Domain Adaptive Diffusion Policy) is a domain-adaptive framework designed to address robotic generalization challenges caused by varying environmental dynamics. It uses Lagged Context Dynamical Prediction to extract disentangled, static environmental features in an unsupervised manner, and employs a Diffusion Injection mechanism to incorporate these features directly into the policy generation process, enabling superior zero-shot adaptation in unseen environments.

**Compliance With Llm Reviewing Policy:**

Affirmed.

**Final Justification:**

The rebuttal has improved my confidence in the paper’s technical contribution and positioning.

**Key Questions For Authors:**

* Does the lag step $\Delta t$ require manual tuning for tasks with different dynamic scales (e.g., high-speed locomotion vs. precise manipulation)? Is there a method to adaptively determine the optimal lag?
* In highly similar domains, can lagged prediction effectively distinguish subtle dynamical differences (e.g., friction coefficients of 0.1 vs. 0.15)?
* If the training set includes diverse and distinct task types, would the extracted static features suffer from cross-task interference or confusion?
* Some recent relevant studies[1-4] appear to be missing. The authors should consider including them and discussing their relationship to the proposed method.

References:

[1] Cheng, S., Ma, L., Chen, Z., Mandlekar, A., Garrett, C.R. and Xu, D., Generalizable Domain Adaptation for Sim-and-Real Policy Co-Training. In The Thirty-ninth Annual Conference on Neural Information Processing Systems, 2025

[2] Liu, T., Li, J., Zheng, Y., Niu, H., Lan, Y., Xu, X. and Zhan, X., Skill Expansion and Composition in Parameter Space. In The Thirteenth International Conference on Learning Representations, 2025

[3] Pace, M.A., Dan, P., Ning, C., Bhardwaj, A., Du, A., Duan, E.W., Ma, W.C. and Kedia, K., 2025. X-Diffusion: Training Diffusion Policies on Cross-Embodiment Human Demonstrations. arXiv preprint arXiv:2511.04671.

[4] Niu, H., Chen, Q., Liu, T., Li, J., Zhou, G., Zhang, Y., Hu, J. and Zhan, X., xted: Cross-domain adaptation via diffusion-based trajectory editing. arXiv preprint arXiv:2409.08687.

**Limitations:**

* By discarding time-varying information to isolate static features, the model may struggle in POMDP settings where transient cues are essential for state estimation.
* The framework assumes static domain properties and may fail to adapt if environmental parameters shift abruptly during task execution.

**Strengths And Weaknesses:**

Strengths
* Effectively filters out noise to extract pure, static environmental features using the "Lagged Prediction" strategy.

* Uses "Diffusion Injection" instead of simple concatenation, allowing environmental data to deeply guide action generation.

* Significant out-performance in unseen environments (Out-of-Distribution) compared to baseline models.

* Operates entirely via unsupervised learning, eliminating the need for manual labels like friction or gravity coefficients.

Weaknesses
* Performance relies on tuning the lagged time step ($\Delta t$); sub-optimal values can hinder representation learning.
* The multi-step denoising process of diffusion models leads to higher computational costs for real-time control.
* Assumes environmental features remain constant over short periods; it may struggle with rapidly shifting dynamic parameters.

---

> ### Author Rebuttal · Authors · 2026-03-30
>
> # Reply to Reviewer SXQE
>
> We thank Reviewer SXQE for the positive evaluation and the recognition of our key contributions, including "effectively filters out noise to extract pure, static environmental features," the use of "Diffusion Injection instead of simple concatenation," and the "significant out-performance in unseen environments." We address all questions below. We also invite the reviewer to refer to our responses to other reviewers for additional results and discussions that may be of interest.
>
>
> ## W1 / Q1. Delta t Tuning Across Tasks
>
> > *Does the lag step Delta t require manual tuning for tasks with different dynamic scales? Is there a method to adaptively determine the optimal lag?*
>
> **No tuning is needed.** All main results (Table 1) adopt $\Delta t = \infty$, **consistent across all tasks and environments**. This is empirically supported by monotonic improvement in both representation quality (Table 2) and downstream performance (Tables 3, 4) as $\Delta t$ increases. Theoretically, $\Delta t = \infty$ retains only static, time-invariant information, making it a universal and parameter-free default choice.
>
> ---
>
> ## W2. Computational Cost of Diffusion Denoising
>
> > *The multi-step denoising process of diffusion models leads to higher computational costs for real-time control.*
>
> DADP uses only **5 inference steps** with DDIM sampling (Table 6, Appendix B.3), introducing no extra cost during the diffusion process. We further show that DADP's representation-biased prior enables efficient generation even with a single denoising step:
>
> | Environment | Diffusion (5-step) | DADP (5-step) | Diffusion (1-step) | DADP (1-step) |
> |---|---|---|---|---|
> | Walker2d | 3722 | **3991** | 158 ($\downarrow 85.8$%) | **2830 ($\downarrow 29.1$%)** |
> | HalfCheetah | 3509 | **4100** | 504 ($\downarrow 85.6$%) | **1357 ($\downarrow 66.9$%)** |
>
> This demonstrates that the representation-biased prior enables a much more efficient diffusion process.
>
> ---
>
> ## W3 / Q3. Static Feature Assumption / Cross-Task Interference
>
> > *Assumes environmental features remain constant over short periods; it may struggle with rapidly shifting dynamic parameters or cross-task interference.*
>
> Thank you for the thoughtful concern. DADP is designed for settings where domain parameters remain constant within an episode — a standard assumption discussed in Section 6 along with possible extensions.
>
> That said, for non-stationary MDPs, since the online context is always drawn from the most recent history, the encoder can still capture the current dynamics from up-to-date context. We empirically validate this by testing the DADP Walker2d checkpoint under four friction variation schedules:
>
> - Increasing: Uniformly increasing in $[1.32, 2.32]$
> - Decreasing: Uniformly decresing in $[2.28, 1.28]$
> - Random: Resample from $[1.32, 2.28]$ every 50 steps
> - Leaping: Alternate in $\{1.32, 2.28\}$ in order every 50 steps
>
> | Mode | Seen | Increasing | Decreasing | Random | Leaping |
> | --- | --- | --- | --- | --- | --- |
> | DADP | 4100 $\pm$ 85 | 4348 $\pm$ 144 | 4105 $\pm$ 251 | 4194 $\pm$ 129 | 3772 $\pm$ 128 |
>
>
> DADP remains performant across all schedules. Regarding "cross-task interference": one encoder is trained per task set (e.g., Walker domains) and would not be applied across task types. We appreciate this perspective and consider cross-task generalization an interesting direction for future exploration.
>
>
> ---
>
> ## Q2. Fine-Grained Domain Distinction
>
> > *In highly similar domains, can lagged prediction effectively distinguish subtle dynamical differences (e.g., friction coefficients of 0.1 vs. 0.15)?*
>
> **Yes.** As shown in Figure 3 and Table 2, with $\Delta t = \infty$, despite Walker2d environments including friction-only variations (e.g. $\{1.32, 1.72, 2.28\}$ in Task 4, 5, 6), representations from different domains still form clearly separated clusters, achieving near-supervised performance.
>
>
> ---
>
> ## Q4. Missing References
>
> > *Some recent relevant studies [1-4] appear to be missing.*
>
> We thank the reviewer for pointing out the relevant works and we would include and discuss all four in the revised Related Work in the revised PDF due to the rebuttal length limit.
>
>
> ---
>
> ## Limitations: POMDP Settings
>
> > *May struggle in POMDP settings where transient cues are essential for state estimation.*
>
> **We agree.** Discarding time-varying information is a deliberate trade-off for high-quality static domain representations (Section 6). We have follow-up efforts exploring how to extend DADP to retain time-varying information while preserving domain representation quality.
>
> ---
>
> We sincerely thank Reviewer SXQE again for the encouraging assessment and constructive suggestions, especially the recommended references. We hope our responses and additional experiments have addressed your concerns and kindly ask whether the reviewer would consider raising the score in light of our clarifications. We are happy to incorporate any further suggestions.

---

> > ### Author Rebuttal · Reviewer_SXQE · 2026-04-03
> >
> > I feel the rebuttal has substantially improved my confidence in the paper’s technical contribution and positioning. I look forward to seeing the new experiments and discussions of related work in the revised draft. I am planning to raise my score.

---

> > > ### Author Response · Authors · 2026-04-03
> > >
> > > Thanks for raising the score! We will incorporate all the discussed points in the revised version. We really appreciate the reviewer's time and effort in helping us strengthen this work.

---

### Official Review · Reviewer_eVcv · 2026-03-13

**Soundness:** 3
**Presentation:** 3
**Significance:** 3
**Originality:** 3
**Overall Recommendation:** 5
**Confidence:** 4

**Summary:**

This paper tackles the domain adaptive policy learning through the perspective of context representation learning. The core idea is that context representations learned from temporal adjacent history can entangle static domain factors with time-varying information, which hurt zero-shot adaptation. To address this, the authors propose constructing Lagged Context Dynamical Prediction (LCDP) to learn more stable domain representations, and then incorporate these representations into diffusion policy as a part of the prior distribution, and jointly denoising. LCDP is evaluated on locomotion and manipulation benchmarks under both IID and OOD zero-shot adaptation settings. The ablations and visualizations support the separability of domain representation and advantages of injecting context representation as prior and joint denoising.

**Compliance With Llm Reviewing Policy:**

Affirmed.

**Final Justification:**

I am raising my score from 4 to 5. The paper presents a technically clean approach with well-motivated design choices. The rebuttal resolved my two main concerns: the MetaDiffuser comparison is reasonably addressed through ablations covering the key design differences, and the scope limitation to shared rewards is clarified as a deliberate choice with supporting evidence for broader applicability. I recognize this paper as a solid contribution with clear empirical gains and manageable remaining limitations. The additional experiments from other reviewer responses, e.g., OOD evaluation and extended ablations further strengthen confidence in the method.

**Key Questions For Authors:**

* Could the authors include MetaDiffuser in the experiments as baseline? If it is not feasible, could the authors explain clearly why such a comparison can not be made under the current protocol?

* Do authors have idea and thoughts on extending this framework to diffusion planning methods? It seems like a very natural transition.

**Limitations:**

Yes

**Strengths And Weaknesses:**

**Strengths**

* The paper addresses an important problem in learning-based control and meta-RL, which is how to infer domain information from history and use it effectively for policy learning and adaptation.

* The proposed method is technically sound and quite clean. The lagged context construction is simple and well motivated to reduce the confounding from instantaneous information. The design of injecting context representation as a part of prior is natural and more effective than naive conditioning.

* The method works empirically robust on the locomotion and manipulation tasks, under both IID and OOD settings. The ablations demonstrates the effects of lagged context length on extracting domain representation, and learning efficiency of different representation utilization choices.

**Weaknesses**

* The authors position MetaDiffuser as the closed related work, but it is not included in the empirical comparison. The empirical analysis could be improved by including diffusion planning baseline like MetaDiffuser.

* DADP is restricted to domains with shared state/action/reward and different transition dynamics, which can not be used as a general transfer method to any arbitrary unseen tasks.

---

> ### Author Rebuttal · Authors · 2026-03-30
>
> # Reply to Reviewer eVcv
>
> We thank Reviewer eVcv for the positive evaluation and the recognition that our method is "technically sound and quite clean," with "simple and well motivated" lagged context construction and "empirically robust" results. We address all your questions below. We also invite the reviewer to refer to our responses to other reviewers for additional results and discussions that may be of interest.
>
>
> ## W1 / Q1. MetaDiffuser Baseline
>
> > *The authors position MetaDiffuser as the closest related work, but it is not included in the empirical comparison. Could the authors include MetaDiffuser or explain why?*
>
> **We do not include MetaDiffuser mainly because it is not open-sourced.** We have double-checked its project page and confirmed that the code icon is not linked to a GitHub repository.
>
> Apart from the availability issue, we tried our best to compare with MetaDiffuser implicitly through our ablations on the two core differences: (1) representation learning — MetaDiffuser adopts $\Delta t = 1$, while DADP uses $\Delta t = \infty$; (2) representation utilization — MetaDiffuser conditions on the representation in the policy input, while DADP injects it into the prior distribution. Our ablation results (Tables 2-4) support the benefit of both design choices.
>
>
>
>
> ## W2. Limited Scope to Shared State/Action/Reward
>
> > *DADP is restricted to domains with shared state/action/reward and different transition dynamics, which cannot be used as a general transfer method to any arbitrary unseen tasks.*
>
> Thank you for raising this important point. Regarding reward variations, we have provided empirical evidence in **Table 3** (Section 5.3.1): applying a larger $\Delta t$ to Meta-DT yields consistent gains on both domain-adaptive environments (Hopper-Param, Walker-Param) **and** reward-changing environments (Ant-Dir, Cheetah-Dir, Cheetah-Vel), suggesting that the lagged context technique can transfer to reward-variation settings.
>
> Theoretically, DADP's lagged context representation learning and diffusion injection do not inherently require shared rewards. In task sets with reward variations, the different reward functions are likewise characterized by low-dimensional static parameter factors, so the same disentanglement principle should apply.
>
> We acknowledge that the current focus on transition dynamics variation is a deliberate **scope choice**, as it directly associates practical robotics challenges such as sim-to-real transfer and cross-embodiment deployment, where physical parameters (gravity, friction, morphology) differ.
>
> Regarding state/action variation, we appreciate the suggestion and consider it a valuable direction for future exploration.
>
>
> ## Q2. Extension to Diffusion Planning
>
> > *Do authors have ideas and thoughts on extending this framework to diffusion planning methods? It seems like a very natural transition.*
>
> Thank you for the insightful suggestion! We agree this is a very natural extension. We have follow-up efforts exploring how to extend DADP to diffusion planning by rethinking how to steer the diffusion process with domain representations.
>
> ---
>
> We sincerely thank Reviewer eVcv again for the thoughtful review and encouraging feedback. We hope our responses have adequately addressed your concerns regarding the MetaDiffuser comparison and scope limitations. Given the reviewer's recognition of the technical soundness, clean design, and robust empirical results, we kindly ask whether the reviewer would consider raising the score in light of our clarifications. We are happy to incorporate any further suggestions!

---

> > ### Author Rebuttal · Reviewer_eVcv · 2026-04-04
> >
> > Thanks to the authors for the clear and thorough rebuttal. The responses address my main concerns, and I am planning to raise my score accordingly.

---

> > > ### Author Response · Authors · 2026-04-04
> > >
> > > Thanks for raising the score! We will incorporate all the discussed points in the revised version. We really appreciate the reviewer's time and effort in helping us strengthen this work.

---

### Decision · Program_Chairs · 2026-04-30

**Decision:**

Accept (regular)

**Comment:**

This is a paper with solid contributions. All the reviewers support acceptance, with all concerns marked as fully or partially resolved after rebuttal. The reviewers agree that the paper addresses an important problem in domain-adaptive policy learning and that the proposed method is technically clean with well-motivated design choices. The key strengths highlighted include the novelty of injecting domain representations directly into the diffusion process rather than relying on simple concatenation, the strong and consistent empirical gains including out-of-distribution generalization, and the fully unsupervised nature of the approach. The authors were responsive throughout, providing true out-of-support OOD evaluations, extended ablations across all environments, and direct diagnostics that collectively strengthened the paper. I recommend acceptance and encourage the authors to incorporate the additional results, clarified terminology, and related work discussion into the revised draft.